

# Revealing joint evolutions and causal interactions in complex eco-hydrological systems by a network-based framework

Lu Wang[1], Yue-Ping Xu[1*], Haiting Gu[1], Li Liu[1], Xiao Liang[1], Siwei Chen[1]

[1]Institute of Water Science and Engineering, Zhejiang University, Hangzhou, 310058, China

*Correspondence to:* Yue-Ping Xu (yuepingxu@zju.edu.cn)

**Abstract:** Climate change and human activities have evidence to change eco-hydrological systems, yet the complex relationships among ecological (normalized difference vegetation index, gross primary productivity, and water use efficiency) and hydrological variables (runoff, soil water storage, groundwater storage, etc.) remain understudied. This study develops a novel framework based on network analysis alongside satellite data and in-situ observations to delineate the joint evolutions

(phenomena) and causal interactions (mechanisms) in complex systems. The former employs correlations and the latter uses physically constrained causality analysis to construct network relationships. This framework is applied to the Yellow River basin, a region undergoing profound eco-hydrological variations. Results suggest that joint evolutions are controlled by compound drivers and direct causality. Different types of network relationships are found, namely, joint evolution with weak causality, joint evolution with high causality, and asynchronous evolution with high causality. The upstream alpine subregions,

for example, where the ecological subsystem is more influenced by temperature while the hydrological one is more driven by precipitation, show relatively high synchronization but with weak and lagged causality between two subsystems. On the other hand, eco-hydrological causality can be masked by intensive human activities (revegetation, water withdrawals, and reservoir regulation), leading to distinct evolution trends. Other mechanisms can also be deduced. Reductions in growing season water use efficiency are directly caused by the control of evapotranspiration, and the strength of control decreases with the greening

land surface in some subregions. Overall, the proposed framework provides insight into the complex interactions within the eco-hydrological systems for the Yellow River basin and has applicability to broader geographical contexts.

## 1. Introduction

Eco-hydrological systems are complex with various interactions occurring between and within the subsystems of atmosphere, vegetation, soil, and water bodies (Pappas et al., 2017; Yan et al., 2023). These interactions involve forcing and

feedback with intensifying or mitigating mechanisms, causing changes in a single variable to propagate through the entire system. These interactions together dictate a collective behavior (Goodwell et al., 2018). In the context of climate change and increasing human activities, eco-hydrological processes have undergone substantial changes. Therefore, there is a pressing need for a comprehensive understanding of how these systems evolve and the interactions that drive their evolutions. This




suggests the need to identify evolution trends and quantify multivariate dependencies at the system level (including water

components, vegetation growth/productivity, ecosystem water use, etc.).

A comprehensive understanding of a system requires finding the patterns and associations within it as much as possible, which is a major challenge (Runge et al., 2017). Network analysis is a powerful tool to study the relationships between elements in complex systems with a clear visualization (Watts and Strogatz, 1998; Barabási and Albert, 1999). This approach generates undirected or directed networks, where links between pairwise variables are assigned varying weights (typically measured by

correlations). Such weights between variables are often used as a proxy to deduce the underlying physical relationships, which can be either direct or indirect. Recently, network analysis has received growing attention in the field of hydrology, primarily for identifying hydrologically homogeneous sites or basins based on spatial precipitation and streamflow networks (e.g., Sivakumar and Woldemeskel, 2014; Jha et al., 2015; Fang et al., 2017; Yasmin and Sivakumar, 2018) and for analyzing temporal co-occurrence of hydrological extreme events such as floods and droughts (e.g., Boers et al., 2013; Han et al. 2020;

Brunner and Gilleland, 2021; Mondal and Mishra, 2021; Fan et al., 2022; Liu et al., 2022). However, beyond spatial network analysis, this methodology can also be applied to other types of systems, such as exploring relationships among multiple hydrological, meteorological, and ecological variables in a certain region (Goodwell et al., 2018; Jiang and Kumar, 2019; Terán et al., 2023). Recent advancements in ground-based data, remote sensing data, and outputs from various Earth system models provide unprecedented opportunities to simultaneously characterize complex process dynamics across different scales.

In the literature, correlation relationships remain prevalent for modeling eco-hydrological systems in the form of networks (Chauhan and Ghosh, 2020; Runge et al., 2023). In this study, these networks are referred to as correlation-based networks. Correlation is useful for measuring the scalar similarity in dynamic behaviors among variables (Aslam, 2015; Su et al., 2023). However, networks defined solely based on correlations cannot infer causal relationships (Altman and Krzywinski, 2015; Yasmin and Sivakumar, 2018). While eco-hydrological interactions are inherently causal, as changes in one variable are caused

by changes in other system variables (Jiang and Kumar, 2019). Additionally, information on the directionality and lagged effects is also useful (Chen et al., 2024). Causal detection has been proven to enhance the understanding of physical mechanisms and contribute to improved model construction (Wang et al., 2018a). To capture causal interdependencies within the system, causal inference techniques are essential. Obtained causal links can form causality-based networks, which is beneficial to discover the path followed by a perturbation introduced in an eco-hydrological variable.

In recent decades, theories and algorithms for causal inference based on observations have been developed, including Structural Causal Modelling (SCM; Peters et al., 2017), Transfer Entropy (TE; Schreiber, 2000), Graph-based methods (e.g., PC algorithm and Bayesian network; Pearl, 1988; Darwiche, 2009; Dechter, 2013), Granger causality (GC; Granger, 1969), and Convergent Cross Mapping (CCM; Sugihara et al., 2012). These methods have also been applied in hydrology research in several studies. For instance, Jiang and Kumar (2019) used an information flow-based method to investigate the information





flows in a long-memory observed stream chemistry dynamics. Singh and Borrok (2019) conducted the Granger causality

analysis to identify the causes of groundwater patterns. Shi et al. (2022) used the convergent cross mapping (CCM) method to

study drought propagation. Terán et al. (2023) used Peter Clark's momentary conditional independence framework (PCMCI+)

to investigate drivers of three water-use efficiency indices in Europe.

However, capturing causality remains challenging in handling high-dimensional datasets with limited sample sizes, like

other generic problems. The eco-hydrological system is intricate, highly interconnected, and dynamic, necessitating the

consideration of multiple variables to better depict the system (Su et al., 2023). From a computational and statistical perspective,

this complexity significantly impacts the reliability of statistical inference. Previous studies have noted that causal inference

techniques can encounter issues such as high false-positive rates or low recall rates when identifying causal relationships

(Rinderer et al., 2018; Delforge et al., 2022). In addition, considering confounding factors and feedback loops, the results

should be interpreted cautiously due to potential spurious links (Deyle, et al., 2016; Peng and Susan, 2022). To improve

reliability, hybrid approaches should be developed by reintroducing the physical aspects of the problem to exclude or control

for the risk of physically irrelevant results (Delforge et al., 2022). Causality results may also be context-specific, so that

conclusions may not generalize well to different settings or time periods. Ensuring the robustness and applicability of causal

findings across different conditions is also challenging (Runge et al., 2019a).

In these regards, this study develops a network-based framework that aims to comprehensively improve our understanding

of eco-hydrological systems, from the observed evolutions (phenomena) to the underlying complex causal interactions

(mechanisms). More precisely, a wide range of variables, mainly related to different types of water storage, streamflow,

vegetation growth, and ecosystem functioning are used to represent the characteristics of our systems. Climatic forcings and

human activities are considered as potential drivers outside the system. To capture system-level variations, the evolutionary

dynamics of each variable are linked to form correlation-based networks. The joint evolution modules are then detected by

clustering and network metrics are used to assess the network properties. Subsequently, physically possible and plausible links

between the variables are constructed conceptually to physically constrain the core structure of causality-based networks.

Finally, significant contemporaneous and lagged causal links are portrayed quantitatively. Overall, this study contributes to

the understanding of eco-hydrological processes and extends the network analysis application within the realm of

ecohydrology. The Yellow River Basin (YRB) in China, which has undergone significant changes in eco-hydrological

processes due to climate change and intensifying human activities (Luan et al., 2021; Wang et al., 2021; Yin et al., 2021), is

taken as the study case. Our framework has the potential to be generalized and applied to the analysis in different regions of

the world as well.





## 2. Methodology

The general framework for investigating eco-hydrological systems consists of the following main steps, as shown in Figure 2. Relationships between eco-hydrological processes vary intra-annually, so we focus on the most active growing season (April to September).

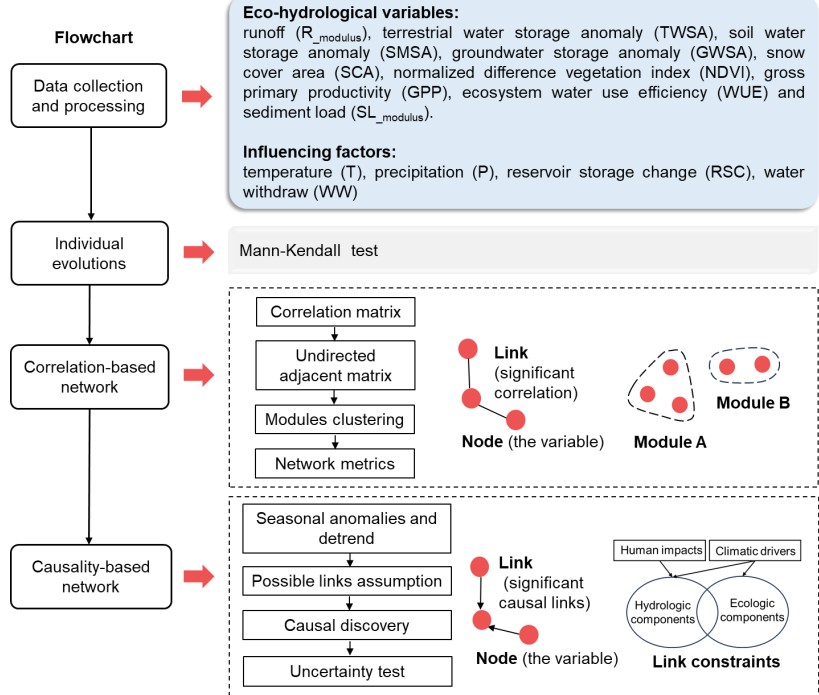

**Figure 1.** Flowchart of the study.

Step I identifies the evolution of each variable using the Mann-Kendall (M-K) test. Thereby, an overview of how eco-hydrological variables change individually is provided.

Step II detects which variables change jointly. Soil water storage (SMSA) and groundwater water storage (GWSA) are two principal components of terrestrial water storage (TWSA), and we therefore remove TWSA to reduce redundant correlations. It is common for gross primary productivity (GPP) and normalized difference vegetation index (NDVI) to be
correlated; however, neither will be excluded because NDVI represents the land surface condition while GPP stands for the photosynthetic activity of the ecosystem. A correlation-based network is constructed for each subregion, and module clustering is employed for the analysis of positive correlations. Modularity as well as the degree of synchronization between hydrological and ecological subsystems are constructed as network metrics.

Step III further investigates the causality between variables. Potential drivers including climatic forcings and human
activities are considered here to fulfill the causal sufficiency. Since multiple variables (more than 10) can generate a large number of causal links with different time lags, some of which may be spurious, empirical knowledge is incorporated into the



causality analysis (Peter Clark momentary conditional independence, PCMCI) to reduce the uncertainty. As the causality can be strongly influenced by the input data, such as the presence of outliers, data length, and the interannual variability of causality, representative subregions are selected to check the robustness of the results.

**2.1 Trend analysis for individual eco-hydrological variables**

The inter-annual trend analysis for eco-hydrological variables is conducted using the commonly applied nonparametric M-K test (Mann, 1945; Kendall, 1948). The positive and negative values of statistic $Z$ indicate the increasing and decreasing tendencies, respectively (Supplementary Material S1). When the absolute value of $Z$ is larger than 1.96, there is a statistically significant trend at the 95% confidence level.

**2.2 Correlation network analysis**

**2.2.1 Construction of the network**

Correlation networks are undirected with no ordering in the nodes defining a link. The nodes represent the eco-hydrological variables and the links between nodes are their correlations. The commonly used Pearson's correlation coefficient (PCC; Pearson, 1895) is used to calculate the strength of connections. Positive PCC indicates a joint evolution between a pair

of variables, while the negative value denotes their opposite evolution trends. PCC is calculated as:

$$PCC(X_i, X_j) = \frac{Cov(X_i, X_j)}{\sqrt{Var(X_i)Var(X_j)}} \tag{1}$$

where $X_i$ and $X_j$ are the time-series data of two variables; $Cov(X_i, X_j)$ is the covariance of $X_i$ and $X_j$; $Var(X_i)$ and $Var(X_j)$ are the variances of $X_i$ and $X_j$, respectively. PCC ranges from -1 to 1, and the correlation is stronger when its absolute value is closer to 1.

Networks for each subregion are constructed using the adjacent matrix $A$, where the links satisfy the significance level of $P<0.05$.

$$A_{ij} = \begin{cases} PCC_{ij}, & if \ P<0.05, \ i \neq j \\ 0, & otherwise \end{cases} \tag{2}$$

where $A_{ij}$ is the weight of the link between variables $i$ and $j$, and is the element of the weighted adjacency matrix $A$. The threshold of $P<0.05$ is regarded to include substantial correlations without excluding too many potential relationships. In

addition, the positively correlated eco-hydrological variables are more densely connected, and we further separate them into several modules using the "cluster walktrap" algorithm in the R package (Pons and Latapy, 2005; Csardi and Nepusz, 2006). The walktrap approach has been widely used and reported to obtain better results on average (Rocha and Filho et al., 2023). Each module represents a group of variables that are more highly correlated among themselves and loosely correlated to others.



### 2.2.2 Network metrics

Modularity ($M$) represents the ability to partition a network into modules, and the modules are detected according to the concentration of links. In this study, this metric is used to measure whether the variables in the system tend to change together or evolve separately. It is defined as

$$m = \frac{\sum_{i,j}\left|A_{ij}\right|}{2} \tag{3}$$

$$M = \frac{\sum_{ij}(A_{ij} - \frac{k_i k_j}{2m})\delta(c_i, c_j)}{2m} \tag{4}$$

where $m$ is the total weighted existing connections; $M$ is the modularity ranging from 0 to 1; $A_{ij}$ is the element of the adjacent matrix; $k_i$ and $k_j$ are the degrees of variable $i$ and variable $j$, respectively; $c_i$ and $c_j$ are the modules of variable $i$ and variable $j$ belonging, respectively. If variables $i$ and $j$ belong to the same module, the function $\delta(c_i, c_j)$ returns 1 and otherwise returns 0 (Newman, 2004).

A new metric $S$ representing the degree of synchronization between hydrological and ecological subsystems is proposed.

It is the ratio of total positive correlations to all the potential links between ecological and hydrological variables:

$$S = \frac{\sum_{ij} A_{ij} \delta^{'}(hs, es)}{2p * q} \tag{5}$$

where $p$ is the number of variables in the hydrological subsystem; $q$ is the number of variables in the ecological subsystem. $hs$ represents the hydrological subsystem and $es$ represents the ecological subsystem, respectively. $\delta^{'}(hs, es)$ returns 1 when two variables $i$ and $j$ are correlated and are in $hs$ and $es$ respectively; otherwise returns 0.

## 2.3 Causal network analysis

### 2.3.1 Causal discovery method

Causality is estimated based on the PCMCI method (Runge et al., 2019a; Runge et al., 2019b). PCMCI is a graphical-based method for linear and nonlinear causal discovery from multivariate time series datasets. The technique consists of two parts: (1) a modified PC algorithm (Colombo and Maathuis, 2014) to estimate the skeleton of the causal network; and (2) the

momentary conditional independence (MCI) test to examine whether the causal connections exist.

Specifically, in an underlying time-dependent system with $N$ variables $X_t^j \in (X_t^1, ..., X_t^N)$ varying in time $t$, the link $X_{t-\tau}^i \rightarrow X_t^j$ (where $\tau$ is a positive time lag) exists if the lagged variable $X_{t-\tau}^i$ has a significant dependence or predictive power over $X_t^j$ while removing the influence of all other potential variables that affect $X_{t-\tau}^i$ or $X_t^j$, except $X_{t-\tau}^i$. These potential variables are casual parents of variable $X_t^j$, denoted as $P(X_t^j) \subset X_t^- = (X_{t-1}, X_{t-2}, ...,)$. In the PC step, the

preliminary parents $P(X_t^j) = (X_{t-1}, X_{t-2}, ..., X_{t-\tau \max})$ of each variable $X_t^j$ are initialized. The null hypothesis is set that



$X_{t-\tau}^{i}$ and $X_{t}^{j}$ are conditional independence. In the first iteration $p = 0$, unconditional independence tests are conducted and $X_{t-\tau}^{i}$ will be removed from $P(X_{t}^{j})$ if the null hypothesis cannot be rejected at a significance level $\alpha_{pc}$. In each next iteration $p \to p+1$, the preliminary parents are sorted according to their absolute statistic value and then conduct conditional independence tests. After each iteration, irrelevant parents are removed from $P(X_{t}^{j})$, and the algorithm converges if no more conditions can be tested. In this way, conditioning on the parent set of the variable $X_{t}^{j}$ is sufficient to identify spurious links in PCMCI. In the second step, the MCI test uses a much smaller set of conditions (generated in the PC stage) to identify cause links for various time delays. MCI is defined as

$$MCI : X_{t-\tau}^{i} \perp X_{t}^{j} \mid P(X_{t}^{j}) \backslash \{X_{t-\tau}^{i}\}, P(X_{t-\tau}^{i}) \tag{6}$$

Both PC and MCI stages use conditional independence tests to measure the strength and the statistical significance of links. Significance level $\alpha_{pc}$ and maximum time delay $\tau_{max}$ are two parameters governing the allowable amount of false-positive link discovery. The linear test statistic is based on partial correlation (ParCorr) and the non-linear connections can be estimated by Conditional Mutual information using the k-nearest neighbor approach (CMI-knn). For more details about the method, please refer to Runge et al. (2019a, 2020).

Although eco-hydrological relationships are nonlinear, our study uses ParCorr to capture significant links. This is because the nonlinear CMI-knn is unstable in the real case study, especially when the data sample is limited (Delforge et al., 2022). Besides, the CMI-knn test is more likely to miss the effective connections and the linear ParCorr test has been reported to detect small nonlinearities as well (Terán et al., 2023). Furthermore, to avoid the penalty of high dimensionality and to maintain high statistical power in conditional independence tests, the maximum time lag $\tau_{max}$ is set at 3 months (seasonal scale). We believe that this time delay is sufficient to detect the majority of significant cause-effect relationships during the growing season. We set a strict significance level of 99% for both condition selection and condition independence tests.

### 2.3.2 Satisfaction of causal assumptions

Faithfulness, causal Markov condition, causal sufficiency, and stationarity of variables are the main assumptions of PCMCI (Runge et al., 2019a). Causal sufficiency refers to the included variables being sufficient to capture the causal relationships between them. However, it always depends on subjective judgment and is difficult to handle due to no boundary of the system (Chauhan, 2023). We account for common influencing factors while controlling high dimensionality. Climatic forcings, i.e., temperature (T) and precipitation (P), as well as reservoir storage change (RSC), are added as potential influencing factors. The actual evapotranspiration (ET) is also added to fulfill causal processes as it governs the ecosystem water use efficiency (WUE) according to the definition. To satisfy the stationarity assumption, the time series of each variable is masked to the growing season months. The series are further detrended and use seasonal anomalies based on the additive model:





$$X_t = T_t + S_t + a_t \qquad (7)$$

where $X_t$ is the original time series, $T_t$ is the trend, $S_t$ is the seasonality, $a_t$ is the remainder, and $t$ denotes time. We first remove the multi-year monthly mean values to obtain seasonal anomalies. The remaining time series are tested for long-term trends using the M-K test. When the null hypothesis of no trend is rejected at a significance level of 0.05, the linear trend is removed from the time series.

### 2.3.3 Using prior knowledge as physical constraints

Eco-hydrological systems present highly interdependent time series (or functional connections), favoring a high false-positive rate. Uncertainties in causality analysis are therefore minimized with the aid of prior knowledge. Our task is to capture the main causal interactions in the region to interpret the potential joint evolutions of eco-hydrological variables. Physically possible and plausible links between the included variables are first hypothesized as a constrained structure (Figure 2). Then, PCMCI tests possible links and provides the final results as a subset of the total possible network, showing causal links, directions, strengths, and time lags. Links that go beyond our hypothesis are not discussed and can be further investigated in the future.

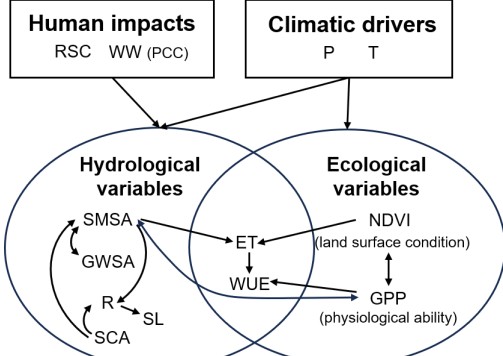

**Figure 2.** The network with physically possible and plausible links between the included variables in the PCMCI analysis. PCMCI will only test shown links for significant causality and yield the final causal network as a subset of this. SMSA= soil moisture storage anomalies; GWSA=groundwater storage anomalies; R=runoff; SL=sediment loads; SCA= snow cover area; NDVI=normalized difference vegetation index; GPP=gross primary productivity; WUE= ecosystem water use efficiency; ET=actual evapotranspiration; P=precipitation; T=temperature; RSC=reservoir storage change; WW=water withdrawals.

We mainly focus on the coupled ecology-hydrology feedbacks that occur at the land surface and climate forcings are considered as external system factors, as are human impacts. The effects of eco-hydrological variables in turn on climatic drivers and human activities are not considered, as it is not the focus of the study. The interactions can be separated into the processes of hydrological→hydrological variables, hydrological→ecological variables, ecological→ecological variables, and ecological→hydrological variables. Some interactions are potentially bidirectional, and the directions of such processes are



not hypothesized. For example, groundwater storage (GWSA) and soil water storage (SMSA) can complement each other. Revegetation measures (NDVI) can improve the sum of GPP, while the physiological ability (GPP) also affects vegetation coverage (NDVI). Besides, water availability (SMSA) facilitates vegetation productivity (GPP) in the short term, while vegetation may in turn influence the SMSA. Note that water withdrawal (WW) is an important anthropogenic influence, but due to the lack of monthly data, the correlation coefficient (PCC) is used to characterize its general association with other

variables in the system.

## 3. Study area and data

### 3.1 Study area

    The Yellow River (Figure 3) is the second-longest river in China (Wang et al., 2020). It originates from the northeastern Qinghai-Tibet Plateau, flowing through the arid and semi-arid Loess Plateau as well as the semi-humid North China Plain, and

finally enters the Bohai Sea. Climatically, the YRB is mainly dominated by the arid and semi-arid continental monsoon with a long-term mean annual PET/P of 2.1 (Xie et al., 2019). Summer serves as the primary rainy season, with precipitation from June to September comprising approximately 70% of the annual total. Due to the vast territory, the annual mean precipitation, potential evaporation, and temperature approximately range from 400 to 570 mm, from 900 to 1050 mm, and from -4 to 14 °C, respectively (Ni et al., 2022).

In this study, the YRB is divided into eight subregions, labelled Region I to Region VIII from the upstream to the downstream (Figure 3 and Table 1). The source region (Region I) is above the Guide (GD) station with a vulnerable alpine eco-environment. The upper reaches include Regions I-IV, covering part of the Qinghai-Tibet Plateau and the driest parts of the Loess Plateau. Note that Region IV is an endorheic area with no runoff. The middle reaches are Regions V-VII between Toudaoguai (TDG) and Huayuankou (HYK) stations, and the lower reaches are Region VIII downstream of HYK. The

dominant land use type in the upper reaches is grassland, while the main types in the middle and lower reaches are cropland and forests (Cao et al., 2022).




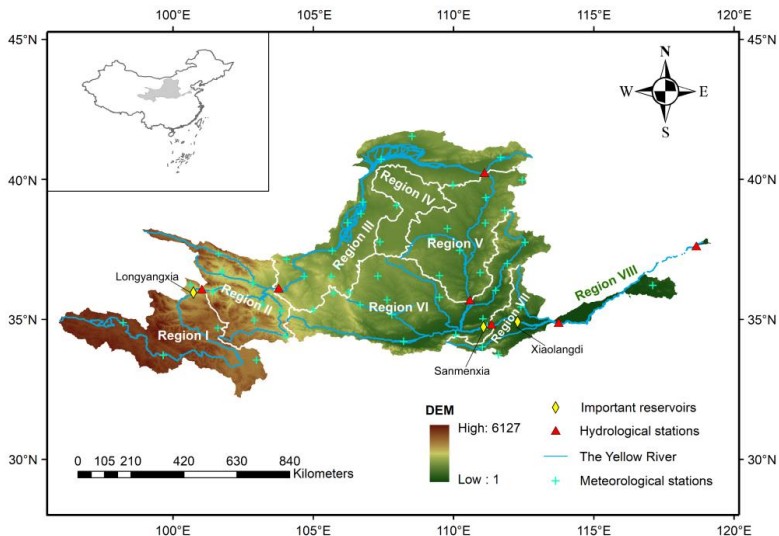

**Figure 3.** Location of the Yellow River Basin and its topography, which is divided into eight subregions based on the secondary basin boundary in China. Distributions of meteorological stations and hydrological stations are also shown in this figure. Region I: Above the Guide; Region II: Guide to Lanzhou; Region III: Lanzhou to Toudaoguai; Region IV: Endorheic Basin; Region V: Toudaoguai to Longmen; Region VI: Longmen to Sanmenxia; Region VII: Sanmenxia to Huayuankou; Region VIII: The downstream of Huayuankou.

Table 1. Summarized information of eight subregions in the YRB

| Region label | Area ($\times 10^4 \, km^2$) | Outflow station | Abbreviation | PET/P | Growing season T (°C) | Location | |
|---|---|---|---|---|---|---|---|
| I | 13.20 | Guide | GD | 1.76 | 7.3 | Qinghai-Tibet Plateau | |
| II | 9.10 | Lanzhou | LZ | 2.05 | 11.7 | Transitional area | Upper reaches |
| III | 15.32 | Toudaoguai | TDG | 3.98 | 18.3 | Loess Plateau | |
| IV | 4.23 | - | - | 3.56 | 18.4 | Loess Plateau | |
| V | 12.24 | Longmen | LM | 2.15 | 18.8 | Loess Plateau | |
| VI | 19.08 | Sanmenxia | SMX | 1.83 | 19.3 | Loess Plateau | Middle reaches |
| VII | 4.17 | Huayuankou | HYK | 1.63 | 21.4 | Transitional area | |
| VIII | 2.24 | Lijin | LJ | 0.88 | 20.0 | The North China Plain | Lower reaches |

Note: PET/P is the long-term dryness index based on Xie et al. (2019). PET is potential evapotranspiration and P is precipitation.



**3.2 Data sources and processing**

**3.2.1 Hydrological data**

The monthly runoff observations from 2003 to 2019 at GD, LZ, TDG, LM, SMX, HYK, and LJ main-stem hydrological stations are collected from the National Hydrological Year Book. Since the Yellow River is one of the most heavily loaded rivers in the world, we collect the sediment loads from the National Hydrological Year Book as well. Gauged streamflow and sediment are not suitable for regional investigation, so we calculate the increments in flow ($R_{modulus}$) and sediment loads

($SL_{modulus}$) for each subregion, i.e., the difference of flow/sediment loads between two gauged stations that are standardized as modulus by area (Xu et al., 2022).

The MODIS-based snow cover product is used to obtain the variation of snow cover area (SCA, Hao et al., 2022) in the source region. Terrestrial water storage (TWS) data are derived from three monthly gridded GRACE products, which are the

GRACE mascon data from the Center for Space Research (CSR, at the University of Texas, Austin) (Save et al., 2016), the GRACE mascon data from Jet Propulsion Laboratory (JPL, at NASA and California Institute of Technology, California) (Swenson and Wahr, 2006; Landerer and Swenson, 2012), and the GRACE mascon data from Goddard Space Flight Center (GSFC, at NASA) (Awange et al., 2011, Luthcke et al., 2017). The three GRACE products are used by taking their ensemble mean values. A few months of data missing during the study period due to "battery management" are interpolated by averaging

the values of adjacent months. All GRACE data used are anomalies relative to a 2004-2009 time-mean baseline, namely, terrestrial water storage anomalies (TWSA). Monthly data simulated by the Noah model of Global Land Data Assimilation System (GLDAS-v2.1; http://disc.sci.gsfc.nasa.gov/services/grads-gds/gldas) are utilized to collect the surface water storage and the soil moisture storage water storage (Xie et al., 2019). Their values are also processed into the anomaly values as surface water storage anomalies (SWSA) and soil moisture storage anomalies (SMSA). Groundwater storage anomalies (GWSA) are

calculated by subtracting SWSA and SMSA from TWSA (Scanlon et al., 2018; Yao et al., 2019).

**3.2.2 Ecological data**

Normalized difference vegetation index (NDVI) and gross primary productivity (GPP) are used as proxies for vegetation growth and photosynthetic activity, respectively. The time series of NDVI at 1 km is obtained from the Terra Moderate Resolution Imaging Spectroradiometer (MODIS) Vegetation Indices Monthly (MOD13A3) product (Didan, 2021)

(https://lpdaac.usgs.gov/products/mod13a3v061/). The GPP dataset is obtained from the MOD17A2H product (Running, et al., 2021a), available at a 500-meter spatial resolution and 8-day temporal resolution. Ecosystem water use efficiency (WUE), measuring the trade-off between carbon gain and water loss (Beer et al., 2007), is quantified as the ratio of GPP to actual evapotranspiration (ET) (Cooley et al., 2022). ET is available from the MOD16A2 product (Running, et al., 2021b) (https://lpdaac.usgs.gov/products/mod16a2v061/). All the time series are processed into monthly scale.



**3.2.3 Auxiliary data**

Monthly average air temperature (T) and precipitation (P) during the period of 2003-2019 at 76 National Meteorological Observatory stations (Figure 3) are derived from the China Meteorological Administration (http://data.cma.cn/). For each subregion, meteorological values are calculated using the Thiessen polygon method based on gauged values. Longyangxia (LYX), Sanmenxia (SMX), and Xiaolangdi (XLD) are important reservoirs engaged in the water-sediment regulation scheme of the Yellow River (Xie et al., 2022). In Region I, we use the data above the reservoir due to concerns within the research community, and we consider reservoir storage changes (RSC) in Regions VI and VII. Water storage changes in the reservoirs are captured based on runoff records from the National Hydrological Year Book. Water withdrawals (WW) data are obtained from the Water Resources Bulletin of the Yellow River (http://www.yellowriver.gov.cn/other/hhgb/). Table S1 is the look-up table for all the data used in this study.

**4. Results**

**4.1. Evolutions of individual variables**

The evolutions of eco-hydrological variables during the growing season across eight subregions are presented in Figures 4(a)-(h). The corresponding M-K test results are plotted in Figure 4(i). The multi-year mean values of the variables are listed in Table S2. As illustrated, the total water storage (TWSA) of the growing season significantly reduced at the basin scale (with a decreasing rate of -5.12 mm yr$^{-1}$) as well as in most of the subregions (Regions III-VIII). Groundwater storage (GWSA) exhibited significant downward trends for the entire basin (at a rate of -6.66 mm yr$^{-1}$) and all the subregions, with depletion increasing from upstream to downstream. However, soil water storage (SMSA) increased at the basin scale (at a rate of 1.56 mm yr$^{-1}$), with a significant upward trend in the source region (Region I) and a downward trend in the lower reaches (Region VIII), respectively. Regarding the regional runoff (R$_{modulus}$), the only subregion passing the M-K test at a 5% significant level is Region VIII, with a substantial decrease over the period 2003-2019. Additionally, the trend in snow cover area in the source region was not significant. However, the snow cover for melting (April) increased, and the onset of melting shifted earlier from June to May (Figure S1). Sediment loads (SL$_{modulus}$) displayed significant evolution trends in Regions VII (where the XLD reservoir is located) and VIII (with severe water withdrawals). Overall, the spatial heterogeneity of hydrological evolutions was generalized as follows: the two upstream regions exhibited increasing trends (significant or insignificant) in all variables, except GWSA; in Regions III-VI, SMSA generally showed insignificant increases while GWSA and TWSA all decreased significantly; and all variables declined in Region VIII.

Ecological trends differed from hydrological trends a lot. Vegetation coverage (NDVI) and productivity (GPP) of the growing season increased by 31.16% and 35.70% for the YRB, respectively. It indicated that the large-scale vegetation

restoration undertaken over the last two decades was effective (Yu et al., 2023). However, the ecosystem water use efficiency

(WUE) of the growing season decreased significantly in most subregions (except in Regions I and VIII) from 2003 to 2019,

even if the annual WUE was generally improved (see Figure S2).

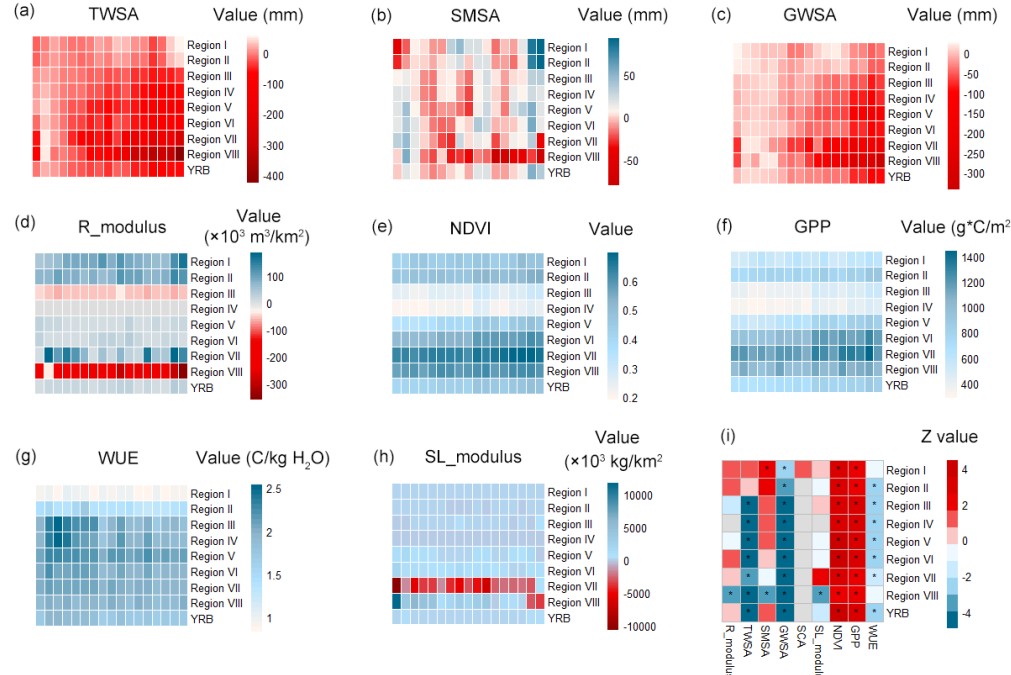

**Figure 4.** Eco-hydrological variables of the growing season, where the horizontal axis represents the year and the vertical axis

is different subregions: (a) terrestrial water storage anomalies (TWSA); (b) soil water storage anomalies (SMSA); (c)

groundwater storage anomalies (GWSA); (d) runoff increment modulus (R $_{modulus}$); (e) normalized difference vegetation index

(NDVI); (f) gross primary productivity (GPP); (g) ecosystem water use efficiency (WUE); (h) sediment load increment

modulus (SL $_{modulus}$); (i) Z statistic values of the M-K test for each eco-hydrological variable. The significance level is taken

as 0.05. A gray box denotes no data; a red box represents a positive trend; a blue box represents a negative trend; the symbol

* means the trend is significant.

**4.2 Correlation-based networks and module detection**

Networks were constructed for each subregion in which evolutions of eco-hydrological variables were linked by

correlations (if significant). The correlations in each network can be found in Figure 5a. GWSA played a significant role in the

formation of negative correlations in most subregions (Figure 5a). Positive correlations were further clustered into different

modules due to their complexity (Figure 5b).





**Figure 5.** (a) Correlation metrics for each subregion; (b) Module composition of positively correlated networks in different subregions. Different gray circles in the background represent different modules. Black lines represent correlations in the same module, and red lines represent correlations in different modules. Blue circles indicate variables of the hydrological subsystem, and green circles indicate variables of the ecological subsystem. WUE is a special ecological indicator represented in yellow circles, as it is the coupling of hydrological (ET) and ecological (GPP) processes.





From Figure 5b, we recognized which parts of the system behaved similarly during the growing season from 2003 to 2019. The modularity was found to be low in the upper two reaches (Regions I and II), meaning that positive correlations were highly connected and were difficult to separate. In particular, the ecological variables in the green circles were correlated with the hydrological variables in the blue circles, and they tightly formed a big module. The $S$ values (synchronization between the two subsystems) for these two regions were 0.25 and 0.37, respectively. This raised the question of whether there was strong feedback between vegetation and water resources that promoted their joint increase. However, the remaining Regions III-VIII had relatively high modularity, ranging from 0.18~0.65. In Regions III and IV, some variables in the ecological (NDVI and GPP) and hydrological (SMSA) subsystems still evolved together, with the decoupling of $R_{modulus}$ and $SL_{modulus}$. The synchronization of the two subsystems was reduced to 0.18 and 0.15, respectively. WUE and GWSA were divided into the same module, both showing downward trends. In Regions V-VIII which were more affected by intensive human activities (Zhang et al., 2023; Yin et al., 2023), the hydrological subsystems in blue and the ecological subsystems in green were found to be decoupled. This indicated that the two subsystems evolved with their own distinct trends. In the downstream of the basin (Region VIII), the modularity decreased on account of reductions in all hydrological components. The mechanisms behind this gradual decoupling correlations of the ecological and hydrological subsystems from the upper to the lower reaches required further investigation.

### 4.3 Causality-based networks

Figure 6 presents all the significant contemporaneous and lagged causal links within the complex eco-hydrological systems. The resulting networks display the drivers of small timescale changes, as the maximum time lag is three months. If a pair of variables exhibit significant causality at multiple time lags and in the same direction, only the strongest lagged link is shown. The most important causal processes typically took place in the current month and with a lag of one month.

#### 4.3.1 Causal links between water components and vegetation

In alpine Regions I-II, NDVI and GPP were found to evolve together with $R_{modulus}$ and SMSA during the growing season positively. Figure 6 uncovered the only weak and lagged causal link between the ecological (green circles) and hydrological subsystems (blue circles), namely, SMSA→GPP at a 1-month lag. It suggested the less water demand for vegetation and the delayed vegetation response to changes in water supply. Instead, increased T (Figure S3) was the dominant factor stimulating GPP, since the contemporaneous T→GPP links with the strengths of 0.78 (Region I) and 0.52 (Region II) were detected. It can be interpreted that these alpine areas are heat-limited and have a certain amount of water resources, resulting in a higher sensitivity of biological photosynthesis, such as carbon allocation and biomass accumulation, to temperature. Meanwhile, increased P (Figure S3) was the crucial driver of the increases in the hydrological subsystem, evidenced by strong





360   contemporaneous and lagged links of P→SMSA, P→R_modulus and P→SL_modulus in the networks. T and P also affected snow

melting (SCA) and further impacted R_modulus positively with a 3-month lag. In general, the exhibiting "joint evolution" between

water components and vegetation were more attributed to their respective drivers instead of direct causality.





**Figure 6.** Causal process networks of eco-hydrological variables in the growing season (April to September) for Regions I-

VIII. A link is only shown if found statistically significant at a 99% confidence level. Link labels in (1), (2) or (3) indicate the

lag at which the connection is found, and only the strongest one is shown in the graph for clarity. (0) means a contemporaneous

link and "—" indicates a contemporaneous link with uncertain direction. All links regarding WW are special, as they are

determined by correlations, marked by PCC. Links between SMSA and NDVI as well as GWSA and GPP are regarded as

spurious ones, which are denoted in dash lines. The red circle under P or (and) T indicates its dominance in controlling the

local eco-hydrological system.

In Regions III-IV, joint evolutions in NDVI, GPP, and SMSA were observed. These are water-limited areas with PET/P

over 3.0 (Table 1), where water availability, rather than heat supply, is the primary factor stimulating the ecological subsystem.

The positive contemporaneous/lagged links of P→GPP and SMSA→GPP were evidence of this. The results indicated a

relatively strong and rapid vegetation response to changes in water supply, and the contribution of vegetation to the

conservation of upper water storage was found (GPP→SMSA with a 1-month lag). Similar links between NDVI and SMSA

were found, but these links could be essentially obtained via the causality between NDVI and GPP together with GPP and

SMSA, and thus we treated them as "spurious" ones. The direct causal interactions and the common driver P mainly contributed

to the joint increases. Compared to Regions I-II, $R_{\_modulus}$ was decoupled from the module. It was found that human water

withdrawals exerted a significant influence on regional runoff (WW→$R_{\_modulus}$), and thus WW was regarded as a significant

contributor to the decoupling of $R_{\_modulus}$ from NDVI, GPP, and SMSA.

In Regions V-VIII, water availability was still important for NDVI and GPP. However, ecological and hydrological

variables evolved in a non-synchronous manner. This could be attributed to the disturbance from human activities. WW

negatively affected GWSA with the magnitudes of -0.81, -0.51, -0.61, and -0.65, respectively. It could further influence soil

water storage via the causality between GWSA and SMSA. WW also decreased $R_{\_modulus}$, that WW→$R_{\_modulus}$ with the

strengths of -0.43 and -0.73 were found in Regions V and VIII, respectively. Reservoir regulation posed strong influences on

$R_{\_modulus}$ and $SL_{\_modulus}$ as well, as strong links with respect to RSC were observed from the networks of Region VI (with SMX

reservoir) and Region VII (with XLD reservoir). These led to great differences between natural and human-induced evolutions,

disrupting the correlations not only between ecological and hydrological variables, but possibly also between hydrological

variables. On the other hand, revegetation measures represented by the "Grain to Green" project have been implemented since

1999 (Zhou et al., 2022). The greening of the land surface (NDVI) contributed to the rapid growth of GPP (strong NDVI→GPP

links). However, excessive vegetation required a lot of extra water to support physiological activities, which had lagged

negative impacts on SMSA (Figure 6). This could also result in different trends in GPP/NDVI and SMSA. A more detailed

insight into this is given in the discussion section.





### 4.3.2 Drivers of WUE variability

As an integrated product of ecological and hydrological processes, WUE was observed to co-evolve with GWSA in

Regions III-VII according to correlation networks, and it remained isolated in the other regions. Conceptually, WUE and

GWSA do not have a direct causality. The causal networks indicated two potential pathways through which GWSA might

indirectly influence WUE, that groundwater could replenish soil water storage, thereby influencing GPP/ET and further

increasing/decreasing the value of WUE. However, the decline in GWSA was also driven by WW, and WUE decreased directly

due to the control of ET (Figure 7).

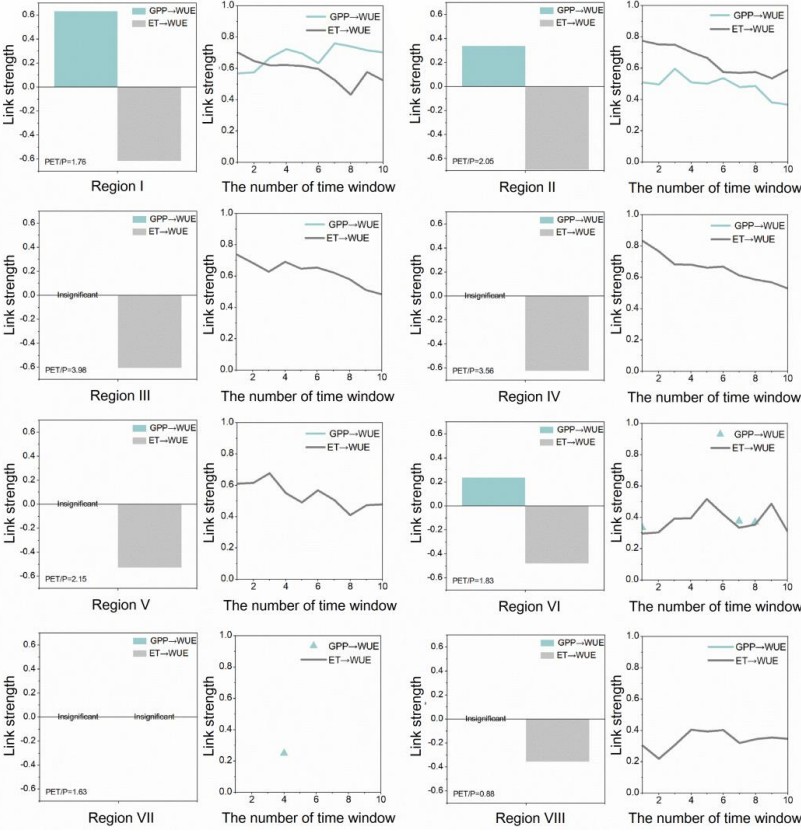

**Figure 7.** Link strengths of GPP versus ET to WUE during the growing season. The bar chart is derived from Figure 6,

representing the overall link strength during 2003-2019. The line graph represents time-varying link strengths (absolute value)

with a sliding window of 8 years.

Either GPP or ET, or both, are directly responsible for WUE changes. In Regions I-II, the control of WUE was exerted

by both GPP and ET. The distinction was that the two types of controls exhibited comparable strengths in Region I (with

insignificant WUE decrease), whereas ET was more dominant in Region II (with significant WUE decrease). In Regions III-

VI (the Loess Plateau), the growing season WUE had a more significant decrease, and causality analysis revealed the control





of ET on WUE. Therefore, the observed decline was attributed to the increases in ET (direct causality), originating from

increased vegetation coverage (NDVI) and soil water storage (SMSA). The increases in NDVI were largely due to afforestation, and changes in SMSA were mainly determined by a combination of vegetation condition, precipitation and groundwater. Generally, the increase in NDVI was more significant than SMSA (Figure 3). The influence of NDVI on ET was also pronounced, with evident contemporaneous and lagged effects (Figure 6). Hence, revegetation contributed significantly to GPP, but it also enhanced ET. As the increase rate of ET exceeded that of GPP, the WUE value was threatened. Interestingly, we

found the control of ET gradually decreased over time in these regions, illustrating that the decreasing trends in WUE were alleviated.

     In addition, one special thing was that the GPP→WUE and ET→WUE links were weak in Regions VII-VIII, particularly in Region VII where the two links were both insignificant. This was due to the high synchronization of monthly ET and GPP (Figure S4), which almost cancelled out their respective contributions to WUE. Consequently, the decreasing trend of WUE

in these two regions was relatively modest.

## 5. Discussion

### 5.1 Network perspective for understanding complex systems

     Correlation network links the individual evolutions of multiple variables in the system, and separate tightly correlated ones into different modules. The joint increases and decreases of variables within and across the subsystems can be therefore

recognized. Proposed network metrics evaluate key aspects of systematic dynamics, including the degree to which the system evolves collectively or independently (the $M$ metric) and the synchronization between subsystems (the $S$ metric). The more complex the system, the more meaningful the approach. However, we discovered that using correlations alone made it difficult to explore the underlying causes. Sometimes intuitively irrelevant variables had similar evolution trends, such as WUE and GWSA, declining together in some cases. Causality is valuable for uncovering underlying mechanisms but has limited

applications in ecohydrology, particularly when multiple variables complicate the disentanglement of cause-and-effect relationships by algorithms. To reduce spurious links, we constructed a conceptual network to constrain the core structure of causal relationships. Theoretically, it is possible to trace the compound causes of changes regarding any variable, contributing to the understanding of eco-hydrological processes. However, we must acknowledge that our study only captured the most important interactions in the watershed and more detailed processes could be too complicated to consider and detect. We found

that joint increases and decreases were controlled by a combination of common drivers, respective drivers, autocorrelation, and causality. Sometimes jointly evolved variables have weak causality (Peng and Susan, 2022). On the other hand, variables can evolve differently but have causality between them. Correlation and causality both make sense, representing phenomena



and mechanisms respectively, while causality-based networks uncover more details. The results revealed by these network approaches are further discussed, taking the YRB as a case study.

### 5.1.1 Climatic forcings can be important to drive joint evolutions

Climatic forcings are critical drivers of variations in the eco-hydrological system, but the effects vary due to heterogeneous characteristics of the subregions. Due to such external drivers, synchronous increases/decreases are ambiguous in terms of mechanism interpretation and may lead to incorrect conclusions. This is especially true for regions where different climatic forcings dominate different eco-hydrological processes but ultimately lead to similar evolutionary trends. Our study presented a good example to illustrate this. The source region of the YRB (Region I) experienced warming and wetter trends in the past decades (Wang et al., 2018b; Yang et al., 2023), and most eco-hydrological variables (except GWSA and WUE) were found to increase jointly. Results showed that the increasing T had a minor influence on the hydrological subsystem, due to the relatively low proportion of snow and glaciers (Table S2). However, T was important for maintaining vegetation growth and physiological activity. P dominated the evolution of hydrological components. Although it could potentially influence the vegetation community indirectly by affecting soil water, the causal interaction between water supply and vegetation was weak. A similar result was also found in Region II, a transitional area between the Tibetan and Loess Plateaus. In the remaining subregions, P was the driver of both hydrological and ecological subsystems. As vegetation activity in these areas depended on water supply, P significantly regulated GPP mainly by influencing soil water for vegetation uptake and was also the main source for replenishing local water resources. In this case, we found joint changes in GPP (NDVI) and SMSA in Regions III and IV with relatively strong causality.

### 5.1.2 Asynchronous evolutions attributed to human activities

Large-scale ecological restoration has been undertaken under the Grain to Green policy, particularly in Regions III-VII. Previous studies have highlighted the negative relationship between water storage and vegetation greenness due to revegetation (e.g. Liu et al., 2023). However, some studies (e.g., Zhang et al., 2022b and Zhou et al., 2022) have challenged this conclusion by showing that a large part of the Loess Plateau has experienced a robust upward trend in surface water yield since the onset of vegetation restoration. We see merit in both conclusions. In our study, soil water storage (SMSA) did not show significant downward trends during 2003-2019 in Regions III-VII (especially in Regions III-V), while groundwater storage (GWSA) in deeper soil layers decreased substantially. On the one hand, the greening of the land surface can contribute positively to soil water storage by allocating more precipitation to infiltration (Lan et al., 2024). We found positive GPP(NDVI)→SMSA effects with a delay of 1 month in Regions III and IV, located in the upper reaches where revegetation was implemented (mainly grassland). That is to say, revegetation leads to water consumption from the soil (Lv et al., 2019; Ge et al., 2020; Li et al.,





2020; Zhao et al., 2022), while it also has the potential to increase the soil water storage in turn. In this case, the overall evolution trends of SMSA and GPP/NDVI were similar. In terms of GWSA, it was consumed by vegetation unidirectionally via SMSA (Figure 6). On the other hand, revegetation was found to have significant adverse impacts on SMSA in Regions V-

VII (Figure 6). This was evidenced by the negative GPP/NDVI→SMSA links with a lag of 3 months, which were more significant than the positive lagged links from GPP to SMSA. These regions are mainly croplands and forests (Cao et al., 2022), where vegetation may have a greater impact on water consumption than grasses due to higher canopy covers and more developed rooting systems (Zhang et al., 2022b). Indirect consumption of deep groundwater storage was also captured, but Region VII was special due to the less replenishment effect between GWSA and SMSA, which might be caused by

groundwater overexploitation and resulting low water levels. Meanwhile, the impact of revegetation on SMSA could further affect $R_{modulus}$ (Chang et al., 2015), although our network did not capture the significant direct link. Overall, revegetation could partly shape the different evolution trends of water components and vegetation indices.

In addition, direct water withdrawal and groundwater exploitation (WW) were reported to significantly influence both surface water and groundwater storage in the middle and lower reaches of the YRB (Yin et al., 2017; Zhang et al., 2023). WW

for irrigation accounted for the majority of the total WW, as YRB is an important agricultural area for wheat and corn (Xie et al., 2019). Our study also quantified the impacts of RSC and WW on regional water storage and runoff. The relevant links all showed strong strength, providing intuitive evidence of anthropogenic influences on decoupled eco-hydrological evolutions in Regions V-VIII.

### 5.2 WUE in the growing season

WUE is an ecological indicator that links physiological processes (GPP) and hydrological processes (ET). Because of this specificity, we did not discuss WUE together with NDVI and GPP and it was considered as an intersection of hydrological and ecological subsystems. Previous studies argued that vegetation restoration on the Loess Plateau led to an improvement in WUE at the annual scale, and the improvement was mainly driven by the increasing GPP (Zhang et al., 2022a; Jiao et al., 2022). Our study found similar upward trends in annual average WUE (Figure S2), but declining growing season WUE was

also detected (Figure 3). Thus, ecosystems may not have adapted well to environmental changes, with reduced functionality and performance in terms of the growing season (Terán et al., 2023). Causality-based networks indicated that the growing season WUE of the YRB was generally controlled by ET (except in Region I), especially in the more arid areas (Regions III-V). This conclusion was consistent with Zhang et al. (2022a). Rapid revegetation increased the amount of GPP; however, such measures also increased the amount of ET. On a positive note, the regulation of ET on WUE showed decreasing trends in many

subregions, especially the ones with relatively low GPP (Figure 7). These trends suggested that the gap between ET and GPP growth rates narrowed as revegetation progressed. It also indicated that the composition of controls on WUE may continue to





change in the future. In addition, as water consumption for carbon uptake varies between vegetation types (Naeem et al., 2023), vegetation structures in the YRB could be further adjusted. To further improve growing season WUE, it is also necessary to minimize the use of water resources by promoting water-saving irrigation systems.

**5.3 Temporal uncertainty of eco-hydrological relationships**

A violin plot is a graphical representation of data distribution. The presence of a flatter or multimodal violin plot indicates a higher degree of uncertainty regarding the causal relationships between variables (Lan et al., 2020). The uncertainty regarding the link strength was characterized using violin plots, given the influence of potential outliers in the series, different data lengths, and the non-stationarity of causality. Due to the requirement of sufficient sample data for causality analysis, sliding

windows of 8-16 years were used to construct different networks and explore the uncertainty of network relationships. Regions I, V and VII were taken as representative cases, standing for the typical alpine area, intensive revegetation area and water-regulated area, respectively.

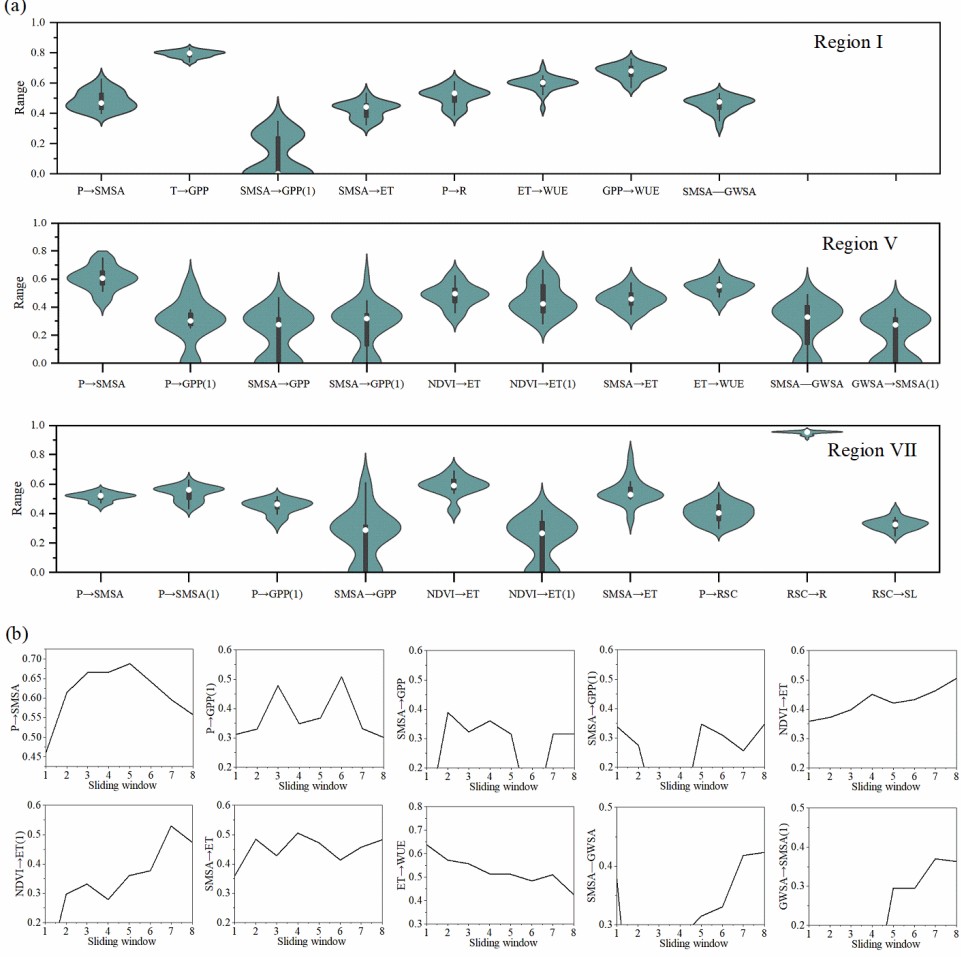





**Figure 8.** (a) Absolute values of causal relationships in three representative subregions of YRB. The important processes of each subregion are selected. If a link is not identified at the significance level, then the link strength is defined as 0. (b) Temporal uncertainty in causal link strength when the time window is 10 years (taking Region V as an example).

In Figure 8, the relative dominance of different eco-hydrological processes (i.e., the median strength represented by the white dots) is generally consistent with the results displayed in Figure 6. This suggests that the relationships found to be significant (Figure 6) are not coincidental, but are generally robust from 2003 to 2019. For example, in Region I, T is more influential to GPP when compared with SMSA, and the strength of GPP→WUE is overall larger than ET→WUE. In contrast to Region I, the link strengths of P→GPP, SMSA→GPP, and NDVI→ET are still stronger in Regions V and VII. RSC is still a vital factor controlling $R_{modulus}$ and $SL_{modulus}$ in Region VII. Nevertheless, some processes show high levels of uncertainty, particularly those with lower link strengths, which may not exhibit significance at some times. Such uncertainty may arise from the random fluctuations of eco-hydrological variables over time, or there may be ongoing evolutionary trends in system relationships. However, the whole study period of 2003-2019 is not a long time, only a small part of the link strengths, e.g., NDVI→ET, NDVI→ET (1), and ET→WUE in Figure 7(b), have obvious trends. Hence, we did not discuss the time-varying network relationships too much in our study.

**5.4 Limitations and future outlooks**

This study adopted a significant amount of remote sensing and reanalysis data, which inevitably resulted in uncertainty in the findings. GRACE data was proven to be a useful tool to reflect the mass changes in TWS of the YRB (Xie, et al., 2019), and we used the ensemble mean values of three products to reduce the uncertainty. NDVI, GPP, and WUE were derived from MODIS products, which were widely used to study the ecological environment of the YRB. For example, Zhang et al. (2022a) explored the spatial-temporal variations of WUE, GPP, and ET utilizing MODIS products across the Ordos Plateau. Liu et al. (2023) checked the accuracy of MODIS-derived GPP data on the Loess Plateau and demonstrated its capacity in ecology applications. SCA was derived from a MODIS-based dataset with good performance. Still, bias errors can be reduced by comparing and fusing data from different datasets in the future.

Although a causality-based network has the potential to clarify relationships between selected variables, depicting all real-world processes can be challenging due to difficulties in data collection, mathematical assumptions, and algorithm performance. Nevertheless, we believe that such findings are important to understand the general watershed functioning and could further guide the development of more accurate and region-specific eco-hydrological models. In this study, we followed the principle of "considering common influencing factors while controlling the high dimensionality", and only the two most important climatic variables (i.e., T and P) were included in causality-based networks. Other climatic factors such as wind speed, radiation, and vapor pressure deficit as well as other eco-hydrological variables such as permafrost index can be



considered in our future work.

## 6. Conclusion

To enhance our understanding of the complex interactions within eco-hydrological systems, including which variables change similarly and potentially why, this study presented a developed framework based on correlation analysis, causality analysis, and a large amount of satellite data and in-situ observations to create network perspectives. The YRB was taken as the study area and the main conclusions were summarized as follows.

Eco-hydrological dynamics in the YRB exhibited significant shifts from 2003 to 2019. During the growing season, TWSA generally decreased, primarily due to GWSA depletion. Meanwhile, NDVI and GPP showed notable increases, whereas WUE declined. Variables in ecological (represented by NDVI and GPP) and hydrological subsystems (represented by $R_{modulus}$, SMSA, etc.) displayed stronger correlations in Regions I-IV (upper reaches) compared to Regions V-VIII (middle and lower reaches). The joint changes in these variables were influenced by common drivers, respective factors, and causality.

Further analysis of causality revealed more detailed interactions within the system. Distinct interaction patterns between ecological and hydrological subsystems emerged across the basin: joint evolution with relatively weak causality (Regions I-II), joint evolution with relatively strong causality (Regions III-IV), and asynchronous evolution with relatively strong causality (Regions V-VIII). We concluded that joint increases observed in Regions I-II primarily resulted from the combined influence of warming and humidifying climate conditions. Whereas in Regions III-IV, joint increases were driven by causality and a common driver P. The divergent trends observed in Regions V-VIII were largely attributed to human activities.

Unexpectedly, a joint decline in growing season WUE and GWSA was observed. The decrease in WUE was primarily regulated by increased ET (direct causality), originating from NDVI and SMSA. GWSA decreased due to WW and the replenishment to SMSA (which further supported GPP and ET). Interestingly, in some subregions, the influence of ET on WUE gradually decreased with the greening of land surface, indicating a mitigation of WUE decline during the growing season. However, optimizing local vegetation structure and water-saving irrigation remain crucial for further improving WUE.

To sum up, this study contributes to the scientific understanding of eco-hydrological systems under a complicated context of climate change and intensive human activities. Furthermore, it demonstrates the potential of causality analysis in revealing complex dependable interactions among multiple variables. The proposed framework not only facilitates the exploration and interpretation of eco-hydrological mechanisms in the YRB, but also holds promise for broader geographical applications.



## Competing interests

At least one of the (co-)authors is a member of the editorial board of Hydrology and Earth System Sciences, and the authors also have no other competing interests to declare.

## Acknowledgements

This study is financially supported by the National Key Research and Development Program of China (Grant No. 2021YFC3201105) and the National Natural Science Foundation of China (Grant No. 52209036). The authors sincerely acknowledged the valuable data provided by the China Meteorological Administration.

## Code and data availability

The GitHub repository contains further information and codes to run the causal discovery framework: https://github.com/jakobrunge/tigramite/. The data that support the findings for analyses are available from the corresponding author upon reasonable request.

## Author contributions

Lu Wang and Yue-ping Xu conceived the idea and designed the study. Lu Wang performed the analysis, prepared the figures, and wrote the manuscript draft. Yue-ping Xu, Haiting Gu, Li Liu, Xiao Liang, and Siwei Chen reviewed and edited the manuscript.

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
