# Peer review of "Revealing joint evolutions and causal interactions in complex ecohydrological systems by a network-based framework"

_Hydrology and Earth System Sciences, 2024_

## Author Comment (AC1)

**Reply to Referee #1**

Dear reviewer,

First of all, we would like to thank you for the time you have spent reviewing our manuscript. We strongly appreciate the comments for further improvements and valuable feedback made. We have carefully addressed the reviewer's comments and suggestions, and revisions have been made in the revised manuscript. Below are our point-by-point responses to the comments in blue text.

Author and Co-Authors

**COMMENTS FROM REVIEWER#1**

The main objective of this paper is to provide a new perspective to analyze eco-hydrological systems based on network approaches. The integrated framework characterized the joint evolution and causal interactions in the complex system at the levels of "phenomena" and "mechanisms", respectively. In particular, I think this study made good attempts to clarify causality between variables of different types (runoff, soil water storage, groundwater storage, normalized difference vegetation index, gross primary productivity, water use efficiency, etc.) by constructing causal networks. The framework was then applied in the Yellow River Basin, China. The results are generally interesting and reasonable. This paper is overall well-structured and well-written.

Despite the proposed framework is promising, the manuscript requires improvements to better illustrate both the methodology and the results sections. In addition, some grammatic errors and figures should be revised. Below are the detailed comments for consideration.

**Reply:** Thank you for your positive evaluation of our manuscript and your suggestions. We have carefully reviewed and revised our manuscript according to your comments.

**Comments in details**

1. Methodology: The flow chart and a large amount of eco-hydrological variables appear abruptly.

Before introducing the flow chart and methods, I suggest adding a concept diagram depicting interactions between the hydrosphere and the biosphere. This diagram should illustrate the eco-hydrological processes in greater detail than Figure 2. Then the authors are suggested to explain why they have chosen these variables (R, TWSA, SMSA, GWSA, NDVI, etc.) for this study.

**Reply:** Thank you for pointing out this issue. The following conceptual figure depicts the eco-hydrological processes in a watershed.

[Figure]

**Figure 1.** Conceptual illustration of the eco-hydrological processes in a watershed. The blue rectangles are related to hydrological processes, the green rectangles are associated with ecological processes and the white rectangles indicate human activities.

Due to the complexity of the processes, we selected some typical variables to characterize the eco-hydrological system, as well as the main influential factors to the system. Our study focuses on coupled ecology-hydrology feedbacks at the land surface so that climate forcings are treated as external factors. The eco-hydrological variables are as follows:

**Hydrological variables:** Regional runoff (R), soil water storage (SMSA), and groundwater storage (GWSA) are chosen as the main hydrological variables. Besides, regional sediment load (SL) is selected since the Yellow River is known for high sediment loads and efforts have been made to address this problem. Additionally, the Yellow River originates in the Tibetan Plateau, which has snow and

glaciers, so we consider the snow cover (SCA). Some more detailed processes, such as infiltration, are not included due to the challenges of accurately quantifying them with the available data sets.

**Ecological variables:** Vegetation coverage and physiological characteristics are mainly taken into account. Three variables, namely, normalized difference vegetation index (NDVI), gross primary productivity (GPP), and ecosystem water use efficiency (WUE) are used to represent vegetation growing condition, carbon uptake condition, and the trade-off between carbon gain and water loss (evapotranspiration, ET) of terrestrial ecosystems, respectively.

**Influencing factors:** The two main climatic factors, i.e. temperature (T) and precipitation (P), as well as the influence of reservoirs (RSC) and water withdrawals (WW), are considered. Data on sunshine duration, wind speed and relative humidity are also available from meteorological stations. However, monthly sunshine duration and monthly relative humidity in YRB are found to be highly correlated with monthly precipitation, and these two variables are not considered due to redundancy. In addition, the influence of wind speed on eco-hydrological processes is insignificant compared to T and P, so this factor is not included.

As suggested, we will add the conceptual illustration and some justification for using these variables in the revised manuscript (Section 2).

2. Line 152: There are many causal inference methods other than PCMCI, such as Convergent Cross Mapping (CCM) and Granger Causality (GC). Can you briefly explain why PCMCI was used in this study?

**Reply:** Thank you for your comments. We fully agree that several methods have been developed over the last few decades for inferring causal relationships from observational data.

**Granger causality** (GC; Granger, 1969) assumes that the cause provides useful information for predicting the effect at future time steps, and any variable in the system can be represented linearly by lagged values of system variables and an error term. This means that two variables occurring at the same time cannot be causally related. Our study is conducted on a monthly time scale (limited by the data obtained) at which many eco-hydrological processes occur below the data's time resolution, and many contemporaneous relationships will be produced. Additionally, multivariate extensions of GC could fail if too many variables are considered (Runge et al., 2019).

**Convergent cross-mapping (CCM)** infers causality between two variables in nonlinear dynamical systems (Sugihara et al., 2012). If variable *X* can be predicted using the reconstructed system based on the time-delay embedding of variable *Y*, then we know that *X* had a causal effect on *Y*. In general, CCM is restricted to strictly deterministic systems and is therefore less suitable for time series that are stochastic in nature. Moreover, a high false-positive rate was reported when using the CCM, explained by CCM's inability to deal with confounding and synchrony (Ombadi et al., 2020; Delforge et al., 2022). CCM does not have the significance assessment for causality as well.

**In this study, PCMCI is employed for the following reasons:** (1) PCMCI addresses the challenges regarding autocorrelated, high-dimensional time series data by first using a condition-selection step (PC) and then applying a momentary conditional independence (MCI) test. (2) In contrast to GC, PCMCI is more efficient, deals with contemporaneous effects, and provides significant causal links with different time delays. (3) Compared to CCM, PCMCI is easier to use (nonparametric tests) and has a significance assessment for causal links (Runge et al., 2019).

As suggested, the explanation of why PCMCI was used in this study will be added in the revised manuscript in brief.

*References:*

*Granger, C. W. J.: Investigating causal relations by econometric models and cross-spectral methods, Econometrica 37, 424-438, 1969.*

*Sugihara, G. et al.: Detecting causality in complex ecosystems, Science, 338, 496-500, 2012.*

*Ombadi, M., et al.: Evaluation of methods for causal discovery in hydrometeorological systems, Water Resources Research, 56, e2020WR027251, 2020.*

*Delforge, D., et al.: Detecting hydrological connectivity using causal inference from time series: synthetic and real karstic case studies, Hydrol. Earth Syst. Sci., 26, 2181-2199, 2022.*

*Runge, J., Bathiany, S., Bollt, E. et al.: Inferring causation from time series in Earth system sciences, Nat. Commun., 10, 2553, 2019.*

3. Line 154: Provide a full name of PC as the term is first appeared.

**Reply:** Thank you for your careful examination. We have corrected in the revised manuscript.

4. Equation (6): The symbol of ⊥ is not clarified.

**Reply:** Here "⊥" denotes independence. We have added more clarification on Equation (6) in the revised manuscript.

5. Line 190: I think a citation is required for the additive model.

**Reply:** Two references of Ombadi et al. (2020) and Poppe et al. (2023) have been added, in which the anomalies of observations are calculated by subtracting the seasonality and removing the linear trend in a similar way.

*References:*

*Ombadi, M., et al.: Evaluation of methods for causal discovery in hydrometeorological systems, Water Resources Research, 56, e2020WR027251, 2020.*

*Poppe Terán, C., Naz, B.S., Graf, A. et al.: Rising water-use efficiency in European grasslands is driven by increased primary production, Commun. Earth Environ., 4, 95, 2023.*

6. Section 2.3.3: This section is interesting, but how such possible links (physical constraints) incorporated to the causality algorithm (PCMCI) is not clear enough. More explanation is needed.

**Reply:** Thank you for your comment. Sometimes, researchers have prior knowledge about the presence or absence of links and their orientations. There are three types of link assumptions, they are: (1) the assumption that the link from variable $X^i$ to $X^j$ at any lag must exist, and its orientation is fixed (Case 1); (2) the assumption that the link from variable $X^i$ to $X^j$ at any lag may exist, and its orientation is fixed (Case 2); and (3) the assumption that the link from variable $X^i$ to $X^j$ at any lag may exist, but its orientation is not specified (the orientation is then given by the time order; Case 3). In our study, we mainly use the second type of assumption to specify the direction of contemporaneous links. Some interactions are potentially bidirectional, in which case the third type of assumption is used.

[Figure]

| | | |
|---|---|---|
| Case 1: | $X^i$ ——lag 1/2/…/$\tau_{max}$——→ $X^j$ | Link(s) must be there and orientation is fixed. |
| Case 2: | $X^i$ ——lag 1/2/…/$\tau_{max}$——→ $X^j$ ? | Link(s) may be there and orientation is fixed. |
| Case 3: | $X^i$ ←——lag 1/2/…/$\tau_{max}$——→ $X^j$ ? | Link(s) may be there and orientation is not specified. |
| Case 4: | $X^i$ ✖ $X^j$ | No link there. |

Based on the research objectives, some physically "impossible" causal links are also assumed to make the results concise and are not tested in the causality analysis. For example, we assume that WUE is influenced by changes in ET or GPP, so the links directly linking factors such as T and P to WUE are considered inappropriate (Case 4). Precipitation does not appear to be able to directly influence groundwater, so we have removed this for the PCMCI test. Such expert knowledge is incorporated by adding the link_assumptions function to the PCMCI algorithm of the Python package tigramite (https://github.com/jakobrunge/tigramite).

As suggested, more explanation on this will be added in the revised manuscript.

7. Lines 230-236: The Yellow River Basin is divided into several subregions, but the general conditions of these regions are not fully introduced. I suggest that more information on this should be presented, in order to help explain the eco-hydrological mechanisms in the following sections and to help the readers easily relate different subregions to their corresponding results.

**Reply:** Thank you for making us notice. In Section 3.1, we have reorganized the sentences and focus more on introducing the general conditions of the subregions. The detailed modifications in the revised manuscript are as follows:

The upper reaches include Regions I-IV, covering part of the Qinghai-Tibet Plateau and the driest parts of the Loess Plateau. The source region (Region I), located above the Guide (GD) station, has a cold and vulnerable eco-environment where the climate is inland alpine semi-humid, generating 35% of the total annual runoff for the entire basin (Zhan et al., 2024). From west to east, altitude gradually decreases, temperature increases and the climate becomes drier. Region II serves as a transition zone between the source (Region I) and the Loess Plateau (Regions III and IV). Regions III and IV are the driest parts of the Yellow River Basin, characterized by low precipitation, high evapotranspiration and

sparse vegetation coverage. The dominant land use type in the upper reaches is grassland (Cao et al., 2022). It should be noted that the Region IV is an endorheic area with no runoff.

The middle reaches are Regions V-VII between Toudaoguai (TDG) and Huayuankou (HYK) stations and the lower reaches are Region VIII downstream of HYK. These areas have a warm-temperate continental monsoon climate (Zhang et al., 2022), with warmer and wetter climatic conditions and better vegetation coverage from Region V to Region VIII. The main land use type types in the middle and lower reaches are cropland and forest. Compared with the upper reaches, these regions have experienced more intensive human activities, including the return of cropland to forests and excessive water withdrawals for large populations, agricultural irrigation and industrial production (Zhou et al, 2024; Xie et al., 2019).

*References:*

*Cao, Y.P., Xie, Z.Y., Woodgate, W., et al.: Ecohydrological decoupling of water storage and vegetation attributed to China's large-scale ecological restoration programs, J. Hydrol., 615, 128651, https://doi.org/10.1016/j.jhydrol.2022.128651, 2022.*

*Xie, J.K., Xu, Y.P., Wang, Y.T., et al.: Influences of climatic variability and human activities on terrestrial water storage variations across the Yellow River basin in the recent decade, J. Hydrol., 579, 124218, https://doi.org/10.1016/j.jhydrol.2019.124218, 2019.*

*Zhan, H., Yu, D.X., Wang. L., et al.: Stronger influences of grassland growth than grassland area on hydrological processes in the source region of the Yellow River, J. Hydrol., 642, 131886, 2024.*

*Zhou, J.L., Liu, Q., Liang, L.Q. et al.: Water constraints enhanced by revegetation while alleviated by increased precipitation on China's water-dominated Loess Plateau, J. Hydrol., 640, 131731, 2024.*

*Zhang, K.Z., Dong, Z.C., Guo, L., et al.: Allocation of flood drainage rights in the middle and lower reaches of the Yellow River based on deep learning and flood resilience, J. Hydrol., 615, 128560, 2022.*

8. Line 242: I would like to change "which is divided" into "with the basin divided".

**Reply:** Thanks. We have revised the text as suggested.

9. Lines 266-268: The ways to calculate surface water storage and soil water storage are not clear. For example, what specific components from GLDAS are included in soil water storage/surface water

storage? Please add some details.

**Reply:** In our study, soil water storage (SMS) is calculated as the total of soil moisture content from four different soil layers (0-10 cm, 10-40 cm, 40-100 cm, and 100-200 cm) simulated by the Noah land surface model. Surface water storage (SWS) contains snow water equivalent and canopy water storage from Noah land surface model, and water stored in reservoirs. More details will be stated in the revised manuscript.

10. Line 268: Revise "soil moisture storage water storage".

**Reply:** Thanks. We have revised the text as suggested.

11. Line 287: Change "withdrawals" to "withdrawal".

**Reply:** Thanks. We have revised the text as suggested.

12. Figures 4 (a)-(h): Labels for horizontal coordinates are missing.

**Reply:** The horizontal axis represents the year from 2003-2019, and we have revised this figure.

13. Figure 5: I think it is better to present the S values (i.e., the synchronization between the two subsystems) in Figure 5b as well.

**Reply:** Thank you for your suggestion. The *S* values have been added in the revised manuscript.

14. Figure 6: This figure is very important, but some of the lines are not clear enough (especially the dash lines). In addition, the resolution of the figure should be improved.

**Reply:** Thank you for the comments. The width of the dash lines has been increased.

15. Line 336: Change "increase" to "increases".

**Reply:** We have revised the text as suggested.

16. Line 407: Change "insignificant WUE decrease" to "an insignificant WUE decrease". Change "significant WUE decrease" to "a significant WUE decrease".

**Reply:** We have revised the text as suggested.

17. Line 435: I know that PCMCI can calculate autocorrelation of each variable when performing causality analysis. However, I do not see any results regarding autocorrelation. I suggest to add more details about this or just remove "autocorrelation" in this sentence.

**Reply:** Thank you for your comments. We did not clearly explain the flow of information between time-series variables in the methodology section, nor did we illustrate the autocorrelation. The following figure is a simple example of the causal process network, where variables may have self-dependency (i.e., autocorrelation) and cross-dependency (i.e., forcings from other variables) with different time lags (Goodwell et al., 2020). For the variable $X^1$ at time $t$ (the brown circle), its variation is driven by $X^1_{t-1}$, $X^2_{t-2}$, and $X^4_{t-1}$. Therefore, we said that "the phenomenon of synchronous increases is controlled by a combination of common drivers, respective drivers, autocorrelation, and causality". Nevertheless, the autocorrelation results of eco-hydrological variables are not important as the study focuses on the interactions between these variables. In this case, we will add some results on this in the Supplementary Material.

[Figure]

*References:*

*Goodwell, A.E., Jiang, P.S., Ruddell, B.L., et al.: Debates—Does Information Theory Provide a New Paradigm for Earth Science? Causality, Interaction, and Feedback, Water Resour. Res., 56, e2019WR024940, https://doi.org/10.1029/2019WR024940, 2020.*

18. Line 534: Change "to understand" to "for understanding".

**Reply:** Thanks! We have revised the text as suggested.

---

## Author Comment (AC2)

**Reply to Referee #2**

Dear reviewer,

We strongly appreciate your positive feedback and constructive comments that help to substantially improve our study. Our detailed responses to each of your suggestions were attached and we hope these would be helpful to solve your concerns. All comments are given in black and responses are shown in blue text.

Author and Co-authors

**COMMENTS FROM REVIEWER #2**

With great interest, I have read and reviewed the manuscript by Wang et al. This manuscript explores the joint evolution and causal interactions within eco-hydrological systems by introducing a comprehensive framework that integrates correlation relationships, causality analysis, together with satellite data and in-situ observations. Eight subregions of the Yellow River Basin (YRB) that I am interested in are used as cases for study. Correlations between ecological and hydrological subsystems are found to be decoupled in downstream areas, with the underlying causes investigated through causality analysis and attributed to various human activities. In addition, factors such as climatic forcing are found to create spurious relationships between eco-hydrological variables.

To my opinion, the study presents a promising framework and provides some interesting insights on eco-hydrological interactions in the YRB. The topic of the paper is timely and relevant to the readership of this journal. My recommendation is to be accepted after the following points are revised.

**Reply:** Thank you again for your time and effort in reviewing our manuscript, as well as for your valuable suggestions for improvement.

**Major comments:**

(1) One critical issue is that the description of some technical terms is difficult to understand, such as modularity and the degree of synchronization (Section 2.2.2). An explanation in the form of a diagram

would make the terms clearer. A schematic diagram of the causal discovery process (Section 2.3) is also suggested to improve the readability of the corresponding texts.

**Reply:** Thank you for pointing this out. Some terms and the methodology regarding causal discovery were not clearly illustrated. **In the revised manuscript, we have added the schematic diagrams describing the network metrics (i.e., modularity and degree of synchronization)** to the flowchart. The improved Figure 1 is shown below:

**Figure 1.** (a) Conceptual diagram of eco-hydrological processes in a basin. (b) Flowchart of the study. Blue circles represent variables of hydrological subsystem and green circles represent variables of ecological subsystem. Blue lines stand for connections between hydrological variables, green lines are connections between ecological variables, and the red lines indicate connections between hydrological and ecological variables.

**In addition, to better illustrate the causal discovery process in Section 2.3, we have added more information to Figure 2. The detailed information is shown below:**

[Figure]

**Figure 2.** Overview of the causality analysis. (a) An example to illustrate causality, where a lagged variable $X^4_{t-1}$ (the brown square) is said to be a cause of $X^1_t$ (the brown circle) if $X^4_{t-1}$ has a significant dependence or predictive power over $X^1_t$ while removing the effect of all other potential variables influencing $X^4_{t-1}$ or $X^1_t$, except $X^4_{t-1}$ (the yellow squares). (b) Four types of assumptions can be used to construct physically possible and plausible links. $\tau_{max}$ means the maximum lag time. (c) Physically possible and plausible links between the included eco-hydrological variables in PCMCI. PCMCI will test shown links for significant causality and yield the final causal network as a subset of this. SMSA=soil moisture storage anomalies; GWSA=groundwater storage anomalies; R=regional runoff; SL= regional sediment loads; SCA= snow cover area; NDVI=normalized difference vegetation index; GPP=gross primary productivity; WUE=ecosystem water use efficiency; ET=actual evapotranspiration; P=precipitation; T=temperature; RSC=reservoir storage change; WW=water withdrawals.

(2) Further analysis and discussion of the results would be good (especially in Sections 5.1.1 and 5.1.2). There are many studies focusing on the ecohydrological processes in the YRB, more comparisons with these studies are suggested to increase the reliability of the results. In addition, are there any other

cases that illustrate such confounding issues in Section 5.1.1?

**Reply:** Thank you for your comments. As suggested, we have added more comparisons with other studies in Sections 5.1.1 and 5.1.2 to make the discussion more thorough. **Section 5.1.1 focuses on the discussion of common drivers such as climatic forcing, and other cases illustrating such confounding issues have been complemented. The revised texts are as follows:**

"Similarly, Bonotto et al. (2022) found that streamflow and groundwater were forced by rainfall and potential evapotranspiration, and hence the identified causal links might be the result of a third (or more) strong common forcing when identifying causal links by using CCM. A synthetic study showed that streamflow and subsurface flow could always exhibit CCM convergence due to the common meteorological forcing (Delforge et al., 2022).

Our study also presented a good example to illustrate this. The source region of the YRB (Region I) experienced a warmer and wetter climate in the past decades (Wang et al., 2018b), and most eco-hydrological variables (except GWSA and WUE) exhibited a joint increase. Results showed that the increasing T had minor influences on the hydrological subsystem. This is due to the relatively small proportion of snow and glaciers (about 6% of the area; Table S2) and the insignificant contribution of the frozen-ground thawing process to soil moisture and runoff during the growing season (Qin et al., 2017; Yang et al., 2023). However, T was important for maintaining vegetation growth and physiological activity, and similar results can also be found in Bo et al. (2022). P dominated the evolution of hydrological components and Li et al. (2024) also reported that P exerted the greatest impact on terrestrial water storage, soil moisture, and snowmelt water in the source region."

**In Section 5.1.2, we focus on discussing the potential impacts of human activities on asynchronous evolution trends in hydrological and ecological subsystems. The revised texts are as follows:**

"The increase in regional P may also lead to increased SMSA, largely due to enhanced land-atmosphere interactions that accelerate local moisture recycling following revegetation (Zhang et al., 2022b). In Regions III and IV (mainly grassland), we found positive GPP (NDVI)→SMSA effects with a delay of 1 month. That is to say, although revegetation leads to water consumption from the soil (Lv et al., 2019; Ge et al., 2020; Li et al., 2020; Zhao et al., 2022), it is potentially beneficial to soil water storage in turn. Wang et al. (2024) also concluded that revegetation had a notably positive impact

on root zone soil moisture and terrestrial water storage in the upstream grasslands. In this case, the overall evolution trends of SMSA and GPP/NDVI showed similar upward trends in these regions.

On the other hand, revegetation was found to have significant adverse impacts on SMSA in Regions V-VII (Figure 6), which was consistent with Cao et al. (2022). This was evidenced by the negative GPP/NDVI→SMSA links with a lag of 3 months, which were more significant than the positive lagged links from GPP to SMSA. These regions are mainly croplands and forests, having a greater impact on water consumption than grasses due to higher canopy covers and more developed rooting systems (Zhang et al., 2022b)."

(3) Although the authors have stated the importance of proposed approaches, more discussion on this is suggested. Eco-hydrological/hydrological models also analyze eco-hydrological interactions, so what are the advantages of your methods over physically-based models? It is promising that "such findings are important to understand the general watershed functioning and could further guide the development of more accurate and region-specific eco-hydrological models (lines 534-535)", and I think it would be better to give more explanations.

**Reply:** Thank you for raising this important point. We fully agree that eco-hydrological models also play a fundamental role in understanding relevant processes of the system or subsystem.

Models are based partly on differential equations representing known processes and partly on semi-empirical relationships representing unknown processes or approximating known processes (Runge et al., 2019). However, models have uncertainties when simulating internal fluxes/states (Kelleher et al., 2017). Sometimes a model may fit the descriptive statistics of the observational data (e.g., GPP) well. Still, the model may not simulate the physical mechanisms affecting GPP well due to multiple model formulations and parameterizations, even if incorrect, may fit the observations equally well. In addition, many models cannot account for human activities, and how to parameterize various human activities in a model is a problem (Tursun et al., 2024).

**Hence, compared to the physically-based models, the advantages of our approach are summarized as follows.** (1) Inferring causal relationships based on observations is more directly linked to physical processes, avoiding the large uncertainties in physical mechanisms raised from model structure deficiencies and equifinality in parameterizations. (2) According to different research

objectives and available data, our approach is more convenient and more flexible in selecting variables and time scales to study. (3) Our approach better incorporates processes that are difficult to consider in eco-hydrological models (e.g. human activities).

**We see causality analysis through our approach as a tool to understand the overall functioning of the watershed and provide complementary information to guide the establishment of eco-hydrological models.** Models contain several formulations based on "causal assumptions" by developers, and the models that are causally similar to observations (i.e., our causality results) may yield more reliable future projections (Runge et al., 2019). For example, in the area where snowmelt contributes significantly to runoff, a snowmelt module considering the accurate influencing time is required in the model. In places where groundwater contributes greatly to the upper soil layers and the water uptake by roots, modules regarding groundwater and soil water movement should be considered carefully. On the other hand, we have to acknowledge that we cannot observe everything, everywhere, or all the time. Therefore, we promote the use of observations and models together in the future to more formally address the perceptions of causality in hydrology. This will allow us to test our assumptions about eco-hydrological interactions and better prepare for a wide range of possible futures. As suggested, the relevant content will be added in the Discussion section in the revised manuscript.

(4) Occasional grammatical errors should be checked and corrected.

**Reply:** Thank you! We will carefully check the text and correct the errors.

**Minor comments:**

(1) Lines 24-27 - It would be better to expand the introduction of eco-hydrological systems and internal interactions with more information.

**Reply:** As suggested, we have added more introduction to this and rephrased the first paragraph. The detailed information is as follows:

"The hydrosphere and biosphere are intrinsically coupled subsystems of the Earth. Hydrological conditions shape the distribution, structure, and function of terrestrial ecosystems, which, in turn, affect the hydrological components via modulations of land-atmosphere water and energy fluxes (Pappas et al., 2017). Hence, eco-hydrological systems are complex with time-dependent interactions occurring

between and within the atmosphere, vegetation, soil, and water bodies (Yan et al., 2023). These interactions contain intensifying and mitigating mechanisms, e.g., vegetation coverage can be enhanced by warmer temperatures, increased water availability, and afforestation, and can be further reduced by the decrease of water storage through root uptake. Together, these interactions among multiple components dictate a collective behavior of the eco-hydrological system (Goodwell et al., 2018). In the context of climate change and increasing human activities, eco-hydrological processes have undergone substantial changes. Therefore, it is a pressing need for a comprehensive understanding of how the system behaves (phenomenon) and unravelling the multivariate interactions (mechanisms) that drive such behaviors at the system level."

(2) Lines 85-87 - More emphasis should be placed on the reason for using the YRB as the study area.

**Reply:** Thanks for your comment. We use the Yellow River basin (YRB) in China due to the following reasons: (1) The YRB is an important ecological corridor, hosting more than 12% of population and creates about 14% of GDP of China. (2) The YRB has a vast area with different climatic conditions, land use types, and human disturbances, providing various types of eco-hydrological regimes for investigation. (3) The YRB has undergone significant changes in eco-hydrological processes due to climate change and intensive human activities. Hence, there is a need to investigate the exhibited evolution trends and the internal mechanisms in such eco-hydrological systems.

As suggested, we will explain the reasons in a brief way in the revised manuscript.

(3) Line 88 - A short presentation of the structure of the paper would be good here.

**Reply:** Thanks for the suggestion. A short presentation of the structure of the paper will be added in the last paragraph of Introduction. The detailed information is as follows:

"The study is structured as follows. Section 2 describes the framework developed. Section 3 introduces the study area and the data used. Section 4 presents the results for each subregion of the YRB, followed by a discussion of the findings in Section 5, including the significance of the study, comparisons with previous studies, and limitations. Finally, some conclusions are drawn in Section 6."

(4) Lines 125-126 - I think the threshold has a significant influence on the construction of the network.

How would the network and clustered modules change if you used a different threshold?

**Reply:** Yes, we fully agree with your comment that the threshold could influence the construction of the network. For comparison, Pearson's correlation coefficient (PCC)>0.4 and PCC>0.5 are also used as thresholds here. Although the existence of some links changes when different thresholds are used, the conclusions of the study remain unchanged. Overall, from the upper to the lower reaches, the modularity (M value) of the synchronous relationships increases (except for the downstream area) and the synchronization between the ecological and hydrological subsystems generally (S value) decreases.

[Figure]

**Figure R1. Synchronous networks and corresponding clustered modules (when PCC>0.4).**

[Figure]

**Figure R2. Synchronous networks and corresponding clustered modules (when PCC>0.5).**

(5) Figure 2 - In terms of physically possible and plausible links, there is a connection between soil water storage (SMSA) and gross primary productivity (GPP), but not between soil water storage (SMSA) and normalized difference vegetation index (NDVI). Why is this?

**Reply:** In this study, NDVI represents vegetation coverage (i.e., the land surface condition) and GPP represents the physiological activity of vegetation. We assume the physically possible and plausible links based on Poppe Terán et al. (2023), i.e., enhanced photosynthesis (GPP) is directly supported by water supply from soil (SMSA) and contributes to the active growth of plants (NDVI). However, increased vegetation coverage (NDVI) in turn consumes water (SMSA) through physiological activity (GPP). Therefore, it is assumed that the relationship between NDVI and SMSA is mediated by the

variable GPP and is considered to be a spurious causal relationship. In the revised manuscript, we will make an explanation for this.

(6) Lines 201-202 - "PCMCI tests possible links and provides the final results as a subset of the total possible network……" However, in Figure 6, there are lines between SMSA and NDVI, as well as between GWSA and GPP (although you have defined them as spurious ones). It seems that the links beyond your hypothesis are also tested. Please check if the expression here is correct.

**Reply:** Thank you for making us notice. Our previous expression was inappropriate.

In this study, we mainly use three types of link assumptions, they are: (1) the link from variable $X^i$ to $X^j$ at some time lags (or contemporaneous) may exist, and its direction is specified; (2) the link from variable $X^i$ to $X^j$ at some time lags may exist, but its direction is not specified (the direction is then given by the time order); and (3) the link that is "physically impossible" to have direct causality and is removed from the test.

As stated in the reply to Question (5), we assume the link between SMSA and NDVI to be spurious due to GPP. Similarly, soil moisture affects the physiological activities of vegetation directly, and we think that groundwater usually affects vegetation by supplying water to the upper soil layers, so here the relationships between GWSA and GPP are assumed to be spurious too. However, we did not remove them from the causality test as "physically impossible" links, but left them in default status as we found such "spurious links" sometimes could be helpful in illustrating eco-hydrological mechanisms. We will revise the description of this in the revised manuscript as suggested.

(7) Section 4.1 - The ecohydrological conditions of eight subregions are not clear enough to me. This may hinder the understanding of the underlying mechanisms in the following sections. Apart from trends, I would recommend describing the average conditions of the subregions in brief.

**Reply:** Thanks for your suggestions. We will clarify the average eco-hydrological conditions of each subregion before describing the trends of eco-hydrological variables. Some sentences will be reorganized to make this section more reader-friendly.

(8) Figure 4 - "A gray box denotes no data ……" However, grey and blue are difficult to distinguish

in Figures 4(d) and 4(h). In addition, Figure 4(h) lacks a ")", and the symbol "*" in Figure 4(i) is difficult to recognize.

**Reply:** Thank you for bringing the point to our attention. We have revised this Figure in the revised manuscript.

(9) Figures 5 and 6 - The resolution needs to be enhanced.

**Reply:** We have enhanced the resolution of these figures.

(10) Figure 6 - This figure is interesting and contains a large amount of information. To my best knowledge, the source region (subregion I) has frozen soil, yet temperature does not appear to significantly affect soil moisture. Could you explain this further?

**Reply:** Thank you for your comments. Seasonally frozen ground, sporadic permafrost, and predominantly continuous permafrost coexist in the source area of the YRB, and the spatial distribution of the frozen ground is diverse and complex (Song et al., 2024). However, it is difficult for us to obtain reliable frozen ground data, so the variable directly describing the frozen ground is not included in this study. The relationship between air temperature (T) and soil water storage (SMSA) may partially reflect the degradation of permafrost due to warmer climate and its potential impact on runoff. However, no significant causality between T and SMSA is captured in our case study, meaning that the increase in SMSA during the growing season (i.e. the thaw period) is mainly due to the contribution of precipitation (P). Similar results are found in Li et al. (2024). Meanwhile, previous studies have indicated that the effects of frozen ground degradation on soil moisture and runoff generally occur in winter in the source area of the YRB (Yang et al., 2023). The reduction in frozen ground depth has only a small positive effect on soil moisture in the growing season (Qin et al., 2017), because the soil ice content decreases and the soil liquid water content increases with increasing temperature, but this water can be quickly consumed by the evapotranspiration process.

As suggested, some explanations will be added in the Discussion of the revised manuscript.

(11) Lines 355-359 - "Instead, increased T (Figure S3) was the dominant factor stimulating GPP……"

"Meanwhile, increased P (Figure S3) was the crucial driver of the increases in the hydrological

subsystem……" To interpret the mechanisms clearer, I prefer to present the temperature and precipitation time series (or trends) in the main body of the manuscript.

**Reply:** Thank you! In the main body of the revised manuscript, we will include the figures on trends in P and T as suggested.

(12) Lines 375-377 - I think "essentially" here is strange. The sentence needs to be rephrased.

**Reply:** Thank you! We have rewritten this sentence in the revised manuscript.

(13) Line 420 - I think "modest" here is not appropriate. The sentence needs to be rephrased.

**Reply:** Thank you for the comment. We have rewritten this sentence in the revised manuscript.

**References:**

Bo, Y., Li, X.K., Liu, K., Wang, S.D., Zhang, H.Y., Gao, X.J., and Zhang, X.Y.: Three Decades of Gross Primary Production (GPP) in China: Variations, Trends, Attributions, and Prediction Inferred from Multiple Datasets and Time Series Modeling, Remote Sens., 14, 2564, https://doi.org/10.3390/rs14112564, 2022.

Bonotto, G., Peterson, T.J., Fowler, K., and Western, A.W.: Identifying Causal Interactions Between Groundwater and Streamflow Using Convergent Cross-Mapping, Water Resour. Res., 58, e2021WR030231, https://doi.org/10.1029/2021WR030231, 2022.

Cao, Y.P., Xie, Z.Y., Woodgate, W., Ma, X.L., Cleverly, J., Pang, Y.J., Qin, F., and Huete, A.: Ecohydrological decoupling of water storage and vegetation attributed to China's large-scale ecological restoration programs, J. Hydrol., 615, 128651, https://doi.org/10.1016/j.jhydrol.2022.128651, 2022.

Delforge, D., de Viron, O., Vanclooster, M., Van Camp, M., and Watlet, A.: Detecting hydrological connectivity using causal inference from time series: synthetic and real karstic case studies, Hydrol. Earth Syst. Sci., 26, 2181–2199, https://doi.org/10.5194/hess-26-2181-2022, 2022.

Ge, J., Pitman, A. J., Guo, W., Zan, B., and Fu, C.: Impact of revegetation of the Loess Plateau of China on the regional growing season water balance, Hydrol. Earth Syst. Sci., 24, 515-533, https://doi.org/10.5194/hess-24-515-2020, 2020.

Kelleher, C., McGlynn, B., and Wagener, T.: Characterizing and reducing equifinality by constraining a distributed catchment model with regional signatures, local observations, and process understanding, Hydrol. Earth Syst. Sci., 21, 3325-3352, https://doi.org/10.5194/hess-21-3325-2017, 2017.

Li, C.C., Zhang, Y.Q., Shen, Y.J., and Yu, Q.: Decadal water storage decrease driven by vegetation changes in the Yellow River Basin, Sci. Bull., 65, 1859-1861, https://doi.org/10.1016/j.scib.2020.07.020, 2020.

Li, X., Zhou, Z.H., Liu, J.J., Xu, C.Y., Xia, J.Q., Wang, P.X., Wang, H., and Jia, Y.W.: Analysis and partitioning of terrestrial water storage in the Yellow River source region, Hydrol. Process., 38, 2, https://doi.org/10.1002/hyp.15097, 2024.

Lv, M.X, Ma, Z.G., Li, M.X., and Zheng, Z.Y.: Quantitative analysis of terrestrial water storage changes under the grain for green program in the Yellow River Basin, J. Geophys. Res. Atmos., 124, 1336-1351, https://doi.org/10.1029/2018JD029113, 2019.

Poppe Terán, C., Naz, B.S., Graf, A. et al.: Rising water-use efficiency in European grasslands is driven by increased primary production, Commun. Earth Environ., 4, 95, https://doi.org/10.1038/s43247-023-00757-x, 2023.

Qin, Y., Yang, D.W., Gao, B., Wang, T.H., Chen, J.S., Chen, Y., Wang, Y.H., and Zheng, G.H.: Impacts of climate warming on the frozen ground and eco-hydrology in the Yellow River source region, China, Sci. Total Environ, 605-606, 830-841, https://doi.org/10.1016/j.scitotenv.2017.06.188, 2017.

Runge, J., Bathiany, S., Bollt, E., et al.: Inferring causation from time series in Earth system sciences, Nat. Commun., 10, 2553, https://doi.org/10.1038/s41467-019-10105-3, 2019.

Song, L., Wang, L., Luo, D. et al.: Assessing hydrothermal changes in the upper Yellow River Basin amidst permafrost degradation, npj Clim. Atmos. Sci., 7, 57, https://doi.org/10.1038/s41612-024-00607-3, 2024.

Tursun, A., Xie, X., Wang, Y.B., Liu, Y., Peng, D.W., and Zheng, B.Y.: Enhancing streamflow simulation in large and human-regulated basins: Long short-term memory with multiscale attributes, J. Hydrol., 630, 130771, https://doi.org/10.1016/j.jhydrol.2024.130771, 2024.

Wang, T.H, Yang, H.B., Yang, D.W., Qin, Y., and Wang, Y.H.: Quantifying the streamflow response to frozen ground degradation in the source region of the Yellow River within the Budyko framework, J. Hydrol., 558, 301-313, https://doi.org/10.1016/j.jhydrol.2018.01.050, 2018b.

Wang, Z.J., Xu, M.Z., Penny, G., Hu, H.C., Zhang, X.P., and Tian, S.M.: Impact of revegetation and agricultural intensification on water storage variation in the Yellow River Basin, J. Hydrol., 635, 131218, https://doi.org/10.1016/j.jhydrol.2024.131218, 2024.

Yang, J.J., Wang, T.H., Yang, D.W., and Yang, Y.T.: Insights into runoff changes in the source region of Yellow River under frozen ground degradation, J. Hydrol., 617, 128892, https://doi.org/10.1016/j.jhydrol.2022.128892, 2023.

Zhang, B.Q., Tian, L., Yang, Y.T., and He, X.G.: Revegetation Does Not Decrease Water Yield in the Loess Plateau of China, Geophys. Res. Lett., 49, e2022GL098025, https://doi.org/10.1029/2022GL098025, 2022b.

Zhao, F.B., Ma, S., Wu, Y.P., Qiu, L.J., Wang, W.K., Lian, Y.Q., Chen, J., and Sivakumar, B.: The role of climate change and vegetation greening on evapotranspiration variation in the Yellow River Basin, China, Agric. For. Meteorol., 316, 108842, https://doi.org/10.1016/j.agrformet.2022.108842, 2022.

---

## Author Response (AR1)

**Response letter**

Dear editor and reviewers,

Thanks again to you and the anonymous reviewers for your valuable comments on our work. We have provided detailed responses to each comment below and revised the manuscript accordingly. We hope that the improved manuscript will satisfy you. For clarity, comments are given in black, our responses are given in blue text, and the revised text in the manuscript are in red.

Best regards,

Lu Wang (on behalf of all authors)

Institute of Water Science and Engineering

Zhejiang University

Hangzhou, 310058, China

**Reply to Referee #1**

The main objective of this paper is to provide a new perspective to analyze eco-hydrological systems based on network approaches. The integrated framework characterized the joint evolution and causal interactions in the complex system at the levels of "phenomena" and "mechanisms", respectively. In particular, I think this study made good attempts to clarify causality between variables of different types (runoff, soil water storage, groundwater storage, normalized difference vegetation index, gross primary productivity, water use efficiency, etc.) by constructing causal networks. The framework was then applied in the Yellow River Basin, China. The results are generally interesting and reasonable. This paper is overall well-structured and well-written.

Despite the proposed framework is promising, the manuscript requires improvements to better illustrate both the methodology and the results sections. In addition, some grammatic errors and figures should be revised. Below are the detailed comments for consideration.

**Reply:** We appreciate the reviewer's positive evaluation and comments on our manuscript. Please find our point-by-point responses below.

**Comments in details**

1. Methodology: The flow chart and a large amount of eco-hydrological variables appear abruptly. Before introducing the flow chart and methods, I suggest adding a concept diagram depicting interactions between the hydrosphere and the biosphere. This diagram should illustrate the eco-hydrological processes in greater detail than Figure 2. Then the authors are suggested to explain why they have chosen these variables (R, TWSA, SMSA, GWSA, NDVI, etc.) for this study.

**Reply:** Thank you for pointing out this issue. The following conceptual figure depicts the eco-hydrological processes in the watershed and has been added in the revised manuscript.

[Figure]

**Figure 1.** Conceptual illustration of the eco-hydrological processes in a watershed. The blue rectangles are related to hydrological processes, the green rectangles are associated with ecological processes and the white rectangles indicate human activities.

Due to the complexity of the processes, we selected some typical variables to characterize the eco-hydrological system, as well as the main influential factors to the system. Our study focuses on ecology-hydrology feedback occurring at the land surface, so climatic forcings are treated as external factors. The selected eco-hydrological variables are as follows:

**Hydrological variables:** Regional runoff ($R_{modulus}$), soil water storage (SMSA), and groundwater storage (GWSA) are chosen as the main hydrological variables. Besides, regional sediment load ($SL_{modulus}$) is selected since the Yellow River is known for high sediment loads and efforts have been made to address this problem. Additionally, the Yellow River originates in the Tibetan Plateau, which has snow and glaciers, so we consider the snow cover (SCA). Some more detailed processes, such as infiltration, are not included due to the difficulty of quantifying them accurately with the available data sets.

**Ecological variables:** Vegetation coverage and physiological characteristics are mainly taken into account. Three variables, namely, normalized difference vegetation index (NDVI), gross primary productivity (GPP), and ecosystem water use efficiency (WUE), are used to represent vegetation

growing condition, carbon uptake condition, and the trade-off between carbon gain and water loss (evapotranspiration, ET) of terrestrial ecosystems, respectively.

**Influencing factors:** The two main climatic factors, i.e. temperature (T) and precipitation (P), as well as the influence of reservoirs (RSC) and water withdrawals (WW), are considered. Data on sunshine duration, wind speed, and relative humidity are also available from meteorological stations. However, monthly sunshine duration and monthly relative humidity in the YRB are found to be highly correlated with monthly precipitation. The two variables are not considered due to the redundancy. In addition, the influence of wind speed on eco-hydrological processes is relatively small when compared to T and P, so this factor is not included as well.

These points have been added in the Methodology section of the revised manuscript (Page 4, Lines 101-110; Page 5, Figure 1a). The detailed revisions are as follows:

"Step I selects variables describing key characteristics/components of the eco-hydrological system and processes the data. Based on Figure 1a, regional runoff ($R_{\text{modulus}}$), terrestrial water storage (TWSA) together with its components (soil moisture storage anomalies, SMSA; groundwater storage anomalies, GWSA) are chosen as the main hydrological variables. Regional sediment load ($SL_{\text{modulus}}$) is also selected since the Yellow River is known for high sediment loads. Besides, snow cover (SCA) of the source region is considered due to its location on the Tibetan Plateau. Vegetation coverage (normalized difference vegetation index, NDVI) and physiological activities (gross primary productivity, GPP) are selected as main ecological variables. In addition, ecosystem water use efficiency (WUE; quantified as the ratio of GPP to actual evapotranspiration) is employed to characterize the trade-off between carbon and water cycles. Due to the difficulty of accurate quantification, more detailed processes such as infiltration and interception are not considered. External climate forcings include precipitation (P) and air temperature (T), and human impacts contain reservoirs (RSC) and human water withdrawals (WW)."

2. Line 152: There are many causal inference methods other than PCMCI, such as Convergent Cross Mapping (CCM) and Granger Causality (GC). Can you briefly explain why PCMCI was used in this study?

**Reply:** Thank you for your comments. We fully agree that several methods have been developed over

the last few decades for inferring causal relationships from observational data.

**Granger causality** (GC; Granger, 1969) assumes that the cause provides useful information for predicting the effect at future time steps, and any variable in the system can be represented linearly by lagged values of system variables and an error term. This means that two variables occurring at the same time cannot be causally related. Our study is conducted on a monthly time scale (limited by the data obtained) at which many eco-hydrological processes occur below the data's time resolution, and many contemporaneous relationships will be produced. Additionally, multivariate extensions of GC could fail if too many variables are considered (Runge et al., 2019).

**Convergent cross-mapping (CCM)** infers causality between two variables in nonlinear dynamical systems (Sugihara et al., 2012). If variable $X$ can be predicted using the reconstructed system based on the time-delay embedding of variable $Y$, then we know that $X$ had a causal effect on $Y$. In general, CCM is restricted to strictly deterministic systems and is therefore less suitable for time series that are stochastic in nature. Moreover, a high false-positive rate was reported when using the CCM, explained by CCM's inability to deal with confounding and synchrony (Ombadi et al., 2020; Delforge et al., 2022). CCM does not have the significance assessment for causality as well.

**In this study, PCMCI is employed for the following reasons:** (1) PCMCI addresses the challenges regarding autocorrelated, high-dimensional time series data by first using a condition-selection step (PC) and then applying a momentary conditional independence (MCI) test. (2) In contrast to GC, PCMCI is more efficient, deals with contemporaneous effects, and provides significant causal links with different time delays. (3) Compared to CCM, PCMCI is easier to use and has the significance assessment for causal links (Runge et al., 2019).

The explanation for the use of PCMCI in this study has been included in the revised manuscript (Page 7, Lines 171-176). The detailed revisions are as follows:

"This method is used because it is able to address the challenges regarding autocorrelated, high-dimensional time series data by first using a condition-selection step (PC; Colombo and Maathuis, 2014) and then applying a momentary conditional independence (MCI) test. Compared to other causal inference methods (such as GC and CCM), PCMCI is more efficient in dealing with high dimensionality, reports significant contemporaneous dependencies, and provides causal relationships with link strengths and different time lags (Runge et al., 2019)."

*References:*

*Granger, C. W. J.: Investigating causal relations by econometric models and cross-spectral methods, Econometrica 37, 424-438, 1969.*

*Sugihara, G. et al.: Detecting causality in complex ecosystems, Science, 338, 496-500, 2012.*

*Ombadi, M., et al.: Evaluation of methods for causal discovery in hydrometeorological systems, Water Resources Research, 56, e2020WR027251, 2020.*

*Delforge, D., et al.: Detecting hydrological connectivity using causal inference from time series: synthetic and real karstic case studies, Hydrol. Earth Syst. Sci., 26, 2181-2199, 2022.*

*Runge, J., Bathiany, S., Bollt, E. et al.: Inferring causation from time series in Earth system sciences, Nat. Commun., 10, 2553, 2019.*

3. Line 154: Provide a full name of PC as the term is first appeared.

**Reply:** Thank you for your careful examination. We have corrected in the revised manuscript.

Related information has been added in Page 2-3, Lines 59-60. The detailed revisions are as follows:

"In recent decades, theories and algorithms for causal inference based on observations have been developed, including Structural Causal Modelling (SCM; Peters et al., 2017), Transfer Entropy (TE; Schreiber, 2000), Graph-based methods such as Peter and Clark's (PC) algorithm and Bayesian network (Pearl, 1988; Darwiche, 2009; Dechter, 2013), Granger causality (GC; Granger, 1969), and Convergent Cross Mapping (CCM; Sugihara et al., 2012)."

4. Equation (6): The symbol of $\perp$ is not clarified.

**Reply:** Here "$\perp$" denotes independence.

The clarification has been presented in the revised manuscript (Page 8, Line 188). The detailed revisions are as follows:

"MCI is defined as

$$MCI : X_{t-\tau}^i \perp X_t^j \mid \widehat{P}(X_t^j) \setminus \{X_{t-\tau}^i\}, \widehat{P}(X_{t-\tau}^i) \tag{6}$$

where $\perp$ denotes (conditional) independence."

5. Line 190: I think a citation is required for the additive model.

**Reply:** Ombadi et al. (2020) and Poppe et al. (2023) have been added, in which the anomalies of observations were calculated by subtracting the seasonality and removing the linear trend in a similar way.

These two references have been added in Section 2.3.2 of the revised manuscript (Page 9, Line 219). The detailed revisions are as follows:

"The series are further detrended and use seasonal anomalies based on the additive model (Ombadi et al., 2020; Terán et al., 2023):

$$X_t = T_t + S_t + a_t \tag{7}$$

where $X_t$ is the original time series, $T_t$ is the trend, $S_t$ is the seasonality, $a_t$ is the remainder, and $t$ denotes time. We first remove the multi-year monthly mean values to obtain seasonal anomalies. The remaining time series are tested for long-term trends using the M-K test. When the null hypothesis of no trend is rejected at a significance level of 0.05, the linear trend is removed from the time series."

*References:*

*Ombadi, M., et al.: Evaluation of methods for causal discovery in hydrometeorological systems, Water Resources Research, 56, e2020WR027251, 2020.*

*Poppe Terán, C., Naz, B.S., Graf, A. et al.: Rising water-use efficiency in European grasslands is driven by increased primary production, Commun. Earth Environ., 4, 95, 2023.*

6. Section 2.3.3: This section is interesting, but how such possible links (physical constraints) incorporated to the causality algorithm (PCMCI) is not clear enough. More explanation is needed.

**Reply:** Thank you for your comment. Sometimes, researchers have prior knowledge about the presence or absence of links and their directions. In our study, there are different types of link assumptions. (1) The assumption that the link from variable $X^i$ to $X^j$ at a lag must exist, and its direction is fixed (Case 1). (2) The assumption that the link from variable $X^i$ to $X^j$ at a lag may exist, and its direction is fixed (Case 2). (3) The assumption that the link from variable $X^i$ to $X^j$ at a lag may exist, but its direction is not specified (the direction is then given by the time order; Case 3). In our study, we mainly use the second type of assumption to specify the direction of contemporaneous links. Some interactions are potentially bidirectional, in which case the third type of assumption is used.

[Figure]

Based on the research objective, some physically "impossible" causal links are also assumed to make the results concise. For example, we assume that WUE is influenced by changes in ET or GPP, so the links directly connecting T (or other factors like P) to WUE are considered inappropriate (Case 4). Precipitation cannot be able to directly influence groundwater, so we have removed this in the PCMCI test. Such knowledge is incorporated by using the link_assumptions function in PCMCI of the Python package (https://github.com/jakobrunge/tigramite).

This point has been rephrased in the revised manuscript (Pages 9-10, Lines 227-234). The detailed revisions are as follows:

"As illustrated in Figure 2b, there are four types of link assumptions and they are: (1) the causal link must exist and its direction is fixed (Case 1); (2) the causal link may exist and its direction is fixed (Case 2); (3) the causal link may exist but its direction is not specified (the direction is then given by the time order; Case 3); and (4) the causal link is physically inappropriate and will not be tested (Case 4). In this study, the second case is designed to specify the direction of potential contemporaneous links, and the third case is used for potential bidirectional interactions. Such knowledge is incorporated by utilizing the link_assumptions function in the Python package tigramite (https://github.com/jakobrunge/tigramite). As a result, physically possible and plausible links between the included variables are hypothesized as a constrained structure (Figure 2c). Then, PCMCI tests possible links and provides the final results as a subset of the total possible network, showing causal links, directions, strengths, and time lags."

7. Lines 230-236: The Yellow River Basin is divided into several subregions, but the general conditions of these regions are not fully introduced. I suggest that more information on this should be presented, in order to help explain the eco-hydrological mechanisms in the following sections and to help the readers

easily relate different subregions to their corresponding results.

**Reply:** Thank you for making us notice. We have reorganized the sentences and introduced more information on the conditions of each subregion.

The added paragraphs can be found in Section 3.1 of the revised manuscript (Pages 10-11; Lines 255-266). The detailed revisions are as follows:

"The upper reaches include Regions I-IV, covering part of the Qinghai-Tibet Plateau and part of the Loess Plateau. The source region (Region I) has a cold and vulnerable eco-environment where the climate is inland alpine semi-humid, generating 35% of the total annual runoff for the entire basin (Zhan et al., 2024). From west to east, the altitude gradually decreases, the temperature rises and the climate becomes drier. Region II is the transitional zone between the source (Region I) and the Loess Plateau (Regions III and IV). Regions III and IV are the driest parts of the YRB, characterized by low precipitation, high evapotranspiration, and sparse vegetation coverage. The dominant land use type in the upper reaches is grassland (Cao et al., 2022).

The middle reaches are Regions V-VII and the lower reaches are Region VIII, with a temperate monsoon climate. From Region V to Region VIII, climatic conditions become warmer and wetter, and vegetation cover increases. The main land use types are cropland and forests. Compared to the upper reaches, these regions have experienced more intensive human activities, including the return of agricultural land to forest and excessive water withdrawals for large populations, agricultural irrigation, and industrial production (Xie et al., 2019; Zhou et al, 2024)."

8. Line 242: I would like to change "which is divided" into "with the basin divided".

**Reply:** Thanks. We have revised the text as suggested.

The modification can be found in Page 11, Line 268. The detailed revisions are as follows:

"**Figure 3.** Location of the Yellow River Basin and its topography, with the basin divided into eight subregions based on the secondary basin boundary in China."

9. Lines 266-268: The ways to calculate surface water storage and soil water storage are not clear. For example, what specific components from GLDAS are included in soil water storage/surface water storage? Please add some details.

**Reply:** In our study, soil water storage (SMS) is calculated as the sum of soil moisture content from four different soil layers (0-10 cm, 10-40 cm, 40-100 cm, and 100-200 cm) simulated by the Noah land surface model. Surface water storage (SWS) includes snow water equivalent and canopy water storage from the Noah land surface model, as well as the water stored in reservoirs and lakes.

The information above has been clarified in Section 3.2.1 of the revised manuscript (Page 13, Lines 299-301). The detailed revisions are as follows:

"SWS contains snow water equivalent and canopy water storage from the Noah model, as well as the volume of water stored in reservoirs and lakes. SMS is calculated as the total soil moisture content from four different soil layers (0-10 cm, 10-40 cm, 40-100 cm, and 100-200 cm)."

10. Line 268: Revise "soil moisture storage water storage".

**Reply:** Thanks. We have revised the text.

The modification can be found in the revised manuscript (Page 12, Line 297-299). The detailed revisions are as follows:

"Monthly data simulated by the Noah model of Global Land Data Assimilation System (GLDAS-v2.1; http://disc.sci.gsfc.nasa.gov/services/grads-gds/gldas) are utilized to collect the surface water storage (SWS) and the soil (moisture) water storage (SMS)."

11. Line 287: Change "withdrawals" to "withdrawal".

**Reply:** Thanks. We have revised the text as suggested.

The modification can be found in the revised manuscript (Page 13, Lines 321-322). The detailed revisions are as follows:

"Water withdrawal (WW) data are obtained from the Water Resources Bulletin of the Yellow River (http://www.yellowriver.gov.cn/other/hhgb/)."

12. Figures 4 (a)-(h): Labels for horizontal coordinates are missing.

**Reply:** The horizontal axis represents the year from 2003-2019, and we have added this information in the figure.

The improved Figure 4 can be found in the revised manuscript (Page 15, Figure 4). The detailed

revisions are as follows:

[Figure]

**Figure 4.** Eco-hydrological variables of the growing season, where the horizontal axis represents the year and the vertical axis is different subregions: (a) terrestrial water storage anomalies (TWSA); (b) soil water storage anomalies (SMSA); (c) groundwater storage anomalies (GWSA); (d) runoff increment modulus (R_modulus); (e) normalized difference vegetation index (NDVI); (f) gross primary productivity (GPP); (g) ecosystem water use efficiency (WUE); (h) sediment load increment modulus (SL_modulus); (i) Z statistic values of the M-K test for each eco-hydrological variable. The significance level is taken as 0.05. A gray box denotes no data; a red box represents a positive trend; a blue box represents a negative trend; the symbol * means the trend is significant.

13. Figure 5: I think it is better to present the S values (i.e., the synchronization between the two subsystems) in Figure 5b as well.

**Reply:** Thank you for your suggestion. The *S* values have been added in the revised manuscript.

The improved Figure 5 can be found in the revised manuscript (Page 16, Figure 5). The detailed

revisions are as follows:

[Figure]

**Figure 5.** (a) Correlation metrics for each subregion; (b) Module composition of positively correlated networks in different subregions. Different gray circles in the background represent different modules.

Black lines represent correlations in the same module, and red lines represent correlations in different modules. Blue circles indicate variables of the hydrological subsystem, and green circles indicate variables of the ecological subsystem. WUE is a special ecological indicator represented in yellow circles, as it is the coupling of hydrological (ET) and ecological (GPP) processes.

14. Figure 6: This figure is very important, but some of the lines are not clear enough (especially the dash lines). In addition, the resolution of the figure should be improved.

**Reply:** Thank you for the comments. The width of the dash lines has been increased.

The improved Figure 6 can be found in the revised manuscript (Page 18, Figure 6). The detailed revisions are as follows:

[Figure]

Region I **High *S* value in correlation relations**

[Figure]

Region III **High *S* value in correlation relations**

[Figure]

Region IV **High *S* value in correlation relations**

[Figure]

Region V **Low *S* value in correlation relations**

[Figure]

Region VI **Low *S* value in correlation relations**

[Figure]

Region VII **Low *S* value in correlation relations**

[Figure]

[Figure]

Region VIII **Low *S* value in correlation relations**

**Figure 6.** Causal process networks of eco-hydrological variables in the growing season (April to September) for Regions I-VIII. A link is only shown if found statistically significant at a 99% confidence level. Link labels in (1), (2) or (3) indicate the lag at which the connection is found, and only the strongest one is shown in the graph for clarity. (0) means a contemporaneous link and "—" indicates a contemporaneous link with uncertain direction. All links regarding WW are special, as they are determined by correlations, marked by PCC. Links between SMSA and NDVI as well as GWSA and GPP are regarded as spurious ones, which are denoted in dash lines. The red circle under P or (and) T indicates its dominance in controlling the local eco-hydrological system.

15. Line 336: Change "increase" to "increases".

**Reply:** We have revised the text as suggested.

The modification can be found in the revised manuscript (Page 17, Line 371). The detailed revisions are as follows:

"This raised the question of whether there was strong feedback between vegetation and water resources that promoted their joint increases."

16. Line 407: Change "insignificant WUE decrease" to "an insignificant WUE decrease". Change "significant WUE decrease" to "a significant WUE decrease".

**Reply:** We have revised the text as suggested.

The modification can be found in the revised manuscript (Page 20, Lines 443-444). The detailed revisions are as follows:

"The distinction was that the two types of controls exhibited comparable strengths in Region I (with an insignificant WUE decrease), whereas ET was more dominant in Region II (with a significant WUE decrease)."

17. Line 435: I know that PCMCI can calculate autocorrelation of each variable when performing causality analysis. However, I do not see any results regarding autocorrelation. I suggest to add more details about this or just remove "autocorrelation" in this sentence.

**Reply:** Thank you for your comments. We did not mention autocorrelations of the variables in the

methodology section. Variables can have self-dependency (i.e., autocorrelation) and cross-dependency (i.e., forcing from other variables) with different time lags (Goodwell et al., 2020). Therefore, we said that "synchronous increases are controlled by a combination of common drivers, respective drivers, autocorrelation, and causality". However, the autocorrelation results of eco-hydrological variables are not important as the study focuses on the interactions between variables. In this case, we have removed "autocorrelation" in this sentence and included related results in the Supplementary Material.

This point has been modified in the revised manuscript (Page 22, Line 470-471; Supplementary Material, Table S3). The detailed revisions are as follows:

"The exhibited joint increases and decreases were found to be controlled by a combination of common drivers, respective drivers, and causality."

**Table S3.** The strength of self-dependency (if significant)

| Variable $i$ | Variable $j$ | Time lag of $i$ | Link type $i$--$j$ | Link value | Variable $i$ | Variable $j$ | Time lag of $i$ | Link type $i$--$j$ | Link value |
|---|---|---|---|---|---|---|---|---|---|
| Region I | | | | | Region V | | | | |
| $R$ | $R$ | 1 | --> | 0.23 | $SMSA$ | $SMSA$ | 1 | --> | 0.49 |
| $SMSA$ | $SMSA$ | 1 | --> | 0.59 | $GWSA$ | $GWSA$ | 1 | --> | 0.41 |
| $GWSA$ | $GWSA$ | 1 | --> | 0.62 | $NDVI$ | $NDVI$ | 1 | --> | 0.53 |
| $NDVI$ | $NDVI$ | 1 | --> | 0.33 | | | | | |
| $GPP$ | $GPP$ | 1 | --> | 0.24 | | | | | |
| $GPP$ | $GPP$ | 2 | --> | 0.24 | | | | | |
| $ET$ | $ET$ | 1 | --> | 0.42 | | | | | |
| Region II | | | | | Region VI | | | | |
| $R$ | $R$ | 1 | --> | 0.33 | $R$ | $R$ | 1 | --> | 0.28 |
| $SL$ | $SL$ | 1 | --> | 0.54 | $SMSA$ | $SMSA$ | 1 | --> | 0.47 |
| $SMSA$ | $SMSA$ | 1 | --> | 0.63 | $GWSA$ | $GWSA$ | 1 | --> | 0.39 |
| $NDVI$ | $NDVI$ | 1 | --> | 0.35 | $NDVI$ | $NDVI$ | 1 | --> | 0.51 |
| $wue$ | $wue$ | 1 | --> | 0.24 | $RSC$ | $RSC$ | 1 | --> | 0.34 |
| $ET$ | $ET$ | 1 | --> | 0.30 | | | | | |
| Region III | | | | | Region VII | | | | |
| $SL$ | $SL$ | 1 | --> | 0.54 | $SMSA$ | $SMSA$ | 1 | --> | 0.33 |
| $SMSA$ | $SMSA$ | 1 | --> | 0.42 | $GWSA$ | $GWSA$ | 1 | --> | 0.44 |
| $GWSA$ | $GWSA$ | 1 | --> | 0.35 | $NDVI$ | $NDVI$ | 1 | --> | 0.30 |
| $NDVI$ | $NDVI$ | 1 | --> | 0.29 | $T$ | $T$ | 1 | --> | 0.20 |
| | | | | | $ET$ | $ET$ | 1 | --> | 0.24 |
| Region IV | | | | | Region VIII | | | | |
| $SMSA$ | $SMSA$ | 1 | --> | 0.54383 | $SMSA$ | $SMSA$ | 1 | --> | 0.39 |
| $GWSA$ | $GWSA$ | 1 | --> | 0.48938 | $GWSA$ | $GWSA$ | 1 | --> | 0.46 |
| $NDVI$ | $NDVI$ | 1 | --> | 0.65776 | $GPP$ | $GPP$ | 1 | --> | 0.30 |

| $ET$ | $ET$ | 1 | --> | 0.38774 | $ET$ | $ET$ | 1 | --> | 0.29 |
|------|------|---|-----|---------|------|------|---|-----|------|

*References:*

*Goodwell, A.E., Jiang, P.S., Ruddell, B.L, et al.: Debates—Does Information Theory Provide a New Paradigm for Earth Science? Causality, Interaction, and Feedback, Water Resour. Res., 56, e2019WR024940, https://doi.org/10.1029/2019WR024940, 2020.*

18. Line 534: Change "to understand" to "for understanding".

**Reply:** Thanks! We have revised the text as suggested.

The modification can be found in the revised manuscript (Page 26, Line 574-576). The detailed revisions are as follows:

"Nevertheless, we believe that our findings are important for understanding the general watershed functioning and could guide the development of more accurate and region-specific eco-hydrological models."

**Reply to Referee #2**

With great interest, I have read and reviewed the manuscript by Wang et al. This manuscript explores the joint evolution and causal interactions within eco-hydrological systems by introducing a comprehensive framework that integrates correlation relationships, causality analysis, together with satellite data and in-situ observations. Eight subregions of the Yellow River Basin (YRB) that I am interested in are used as cases for study. Correlations between ecological and hydrological subsystems are found to be decoupled in downstream areas, with the underlying causes investigated through causality analysis and attributed to various human activities. In addition, factors such as climatic forcing are found to create spurious relationships between eco-hydrological variables.

To my opinion, the study presents a promising framework and provides some interesting insights on eco-hydrological interactions in the YRB. The topic of the paper is timely and relevant to the readership of this journal. My recommendation is to be accepted after the following points are revised.

**Reply:** Thank you for your time and efforts in reviewing our manuscript, as well as for your valuable suggestions for improvement.

**Major comments:**

(1) One critical issue is that the description of some technical terms is difficult to understand, such as modularity and the degree of synchronization (Section 2.2.2). An explanation in the form of a diagram would make the terms clearer. A schematic diagram of the causal discovery process (Section 2.3) is also suggested to improve the readability of the corresponding texts.

**Reply:** Thank you for pointing this out. Some terms and the methodology regarding causal discovery were not clearly illustrated. In the revised manuscript, we have added the schematic diagrams describing the network metrics (i.e., modularity and degree of synchronization) to the flowchart. The corresponding figure has been added in the revised manuscript (Page 5, Figure 1), and is shown below:

[Figure]

**Figure 1.** The general framework for investigating eco-hydrological systems. (a) The conceptual diagram of eco-hydrological processes in a basin. (b) The detailed flowchart. The blue circle denotes the hydrological variable the green circle represents the ecological variable. The blue line stands for the connection between hydrological variables, the green line means the connection between ecological variables, and the red line is the connection between hydrological and ecological variables.

In addition, to better illustrate the causal discovery process in Section 2.3, we have added more information to Figure 2.

The corresponding figure has been added in the revised manuscript (Page 8, Figure 2), and is shown below:

[Figure]

**Figure 2.** Overview of the causal inference method. (a) An example of causality that a lagged variable $X^4_{t-1}$ (the brown square) is said to be a cause of $X^1_t$ (the brown circle) if $X^4_{t-1}$ has a significant dependence or predictive power over $X^1_t$ while removing the effect of all other potential variables influencing $X^4_{t-1}$ or $X^1_t$ (the yellow squares), except $X^4_{t-1}$. (b) Four types of assumptions to construct physically possible and plausible links. τmax means the maximum lag time. (c) The network with physically possible and plausible links between the included variables in the PCMCI analysis. PCMCI will test shown links for significant causality and yield the final causal network as a subset of this. The dashed line represents the causality considered to be spurious, but we do not remove it from the test as Case 4, as it might help illustrate eco-hydrological mechanisms.

(2) Further analysis and discussion of the results would be good (especially in Sections 5.1.1 and 5.1.2). There are many studies focusing on the ecohydrological processes in the YRB, more comparisons with these studies are suggested to increase the reliability of the results. In addition, are there any other cases that illustrate such confounding issues in Section 5.1.1?

**Reply:** Thank you for your comments. As suggested, we have added more comparisons with other studies in Sections 5.1.1 and 5.1.2 to make the discussion more thorough. Section 5.1.1 focuses on the discussion of common drivers such as climatic forcing, and other cases illustrating such confounding issues have been complemented.

These points have been added to the Discussion section of the revised manuscript (Pages 22, Lines 483-494). The detailed revisions are as follows:

"Bonotto et al. (2022) identified relationships between streamflow and groundwater using CCM. They pointed out that streamflow and groundwater were forced by rainfall and potential evapotranspiration, and hence the identified relationships might be the result of a third (or more) strong common forcing. A synthetic study also showed that the common meteorological forcing could always make streamflow and subsurface flow show CCM convergence (Delforge et al., 2022).

Our study presented good examples to illustrate this as well. The source region of the YRB (Region I) experienced a warmer and wetter climate in the past decades (Wang et al., 2018b; Yang et al., 2023), and we found different drivers and influencing pathways ultimately led to synchronized growth of the variables. Results showed that T was important for variables regarding vegetation growth and physiological activity in this subregion. A similar conclusion was also drawn by Bo et al. (2022). P was discovered to dominate the evolutions of hydrological components in the source region, just as Li et al. (2024) reported. However, increasing T had minor influences on the hydrological subsystem. This is due to the relatively small proportion of snow and glaciers (about 6% of the area; Table S2) and the insignificant contribution of the frozen-ground thawing process to soil moisture and runoff during the growing season (Qin et al., 2017; Yang et al., 2023)."

In Section 5.1.2, we focus on discussing the potential impacts of human activities on asynchronous evolution trends in hydrological and ecological subsystems.

These points have been added to the Discussion section of the revised manuscript (Pages 23, Lines 504-520). The detailed revisions are as follows:

"On the one hand, the greening of the land surface can contribute positively to soil water storage by allocating more precipitation to infiltration (Lan et al., 2024). The increase in regional P may also lead to increased SMSA, largely due to enhanced land-atmosphere interactions that accelerate local moisture recycling following revegetation (Zhang et al., 2022b). In Regions III and IV (mainly grassland), we found positive GPP (NDVI)→SMSA effects with a delay of 1 month. That is to say, although revegetation leads to water consumption from the soil (Lv et al., 2019; Ge et al., 2020; Li et al., 2020; Zhao et al., 2022), it is potentially beneficial for soil water storage in turn. Wang et al. (2024)

also concluded that revegetation had a notably positive impact on root zone soil moisture and terrestrial water storage in the upstream grasslands. In this case, the overall evolution trends of SMSA and GPP/NDVI showed similar upward trends in these regions.

On the other hand, revegetation was found to have significant adverse impacts on SMSA in Regions V-VII (Figure 6), which was consistent with Cao et al. (2022). This was evidenced by the negative GPP (NDVI)→SMSA links with a lag of 3 months, which were more significant than the positive lagged links from GPP to SMSA. These regions are mainly croplands and forests, having a greater impact on water consumption than grasses due to higher canopy covers and more developed rooting systems (Zhang et al., 2022b). Indirect consumption of deep groundwater storage was also captured but Region VII was special due to the less replenishment effect between GWSA and SMSA, which might be caused by groundwater overexploitation and resulting low water levels. Therefore, revegetation can, at least in part, lead to different trends in water components and vegetation indices."

(3) Although the authors have stated the importance of proposed approaches, more discussion on this is suggested. Eco-hydrological/hydrological models also analyze eco-hydrological interactions, so what are the advantages of your methods over physically-based models? It is promising that "such findings are important to understand the general watershed functioning and could further guide the development of more accurate and region-specific eco-hydrological models (lines 534-535)", and I think it would be better to give more explanations.

**Reply:** Thank you for raising this important point. We fully agree that eco-hydrological models also play a fundamental role in understanding relevant processes of the system or subsystem.

Models are based partly on differential equations representing known processes and partly on semi-empirical relationships representing unknown processes or approximating known processes (Runge et al., 2019). However, models have uncertainties when simulating internal fluxes/states (Kelleher et al., 2017). Sometimes a model may fit the descriptive statistics of the observational data well. Still, the model may not simulate the physical mechanisms well due to multiple model formulations and parameterizations, even if incorrect, may fit the observations equally well. In addition, many models cannot account for human activities, and how to parameterize various human activities in a model is a problem (Tursun et al., 2024).

**Hence, compared to physically-based models, the advantages of our approach are summarized as follows.** (1) Our approach is more directly linked to physical processes, avoiding the large uncertainties in physical mechanisms raised from model structure deficiencies and equifinality in parameterizations. (2) According to different research objectives and available data, our approach is more convenient and more flexible in selecting variables and time scales to study. (3) Our approach better incorporates processes that are difficult to consider in eco-hydrological models (e.g. human activities).

**We see causality analysis through our approach as a tool to understand the overall functioning of the watershed and provide complementary information to guide the establishment of eco-hydrological models.** Models contain several formulations based on "causal assumptions" by developers, and the models that are causally similar to observations (i.e., our causality results) may yield more reliable future projections (Runge et al., 2019). For example, in the area where snowmelt contributes significantly to runoff, a snowmelt module considering the accurate influencing time is required in the model. In places where groundwater contributes greatly to the upper soil layers and the water uptake by roots, modules regarding groundwater and soil water movement should be considered carefully. On the other hand, we have to acknowledge that we cannot observe everything, everywhere, or all the time. Therefore, we promote the use of observations and models together in the future to more formally address the perceptions of causality in hydrology. This will allow us to test our assumptions about eco-hydrological interactions and better prepare for a wide range of possible futures. The relevant content has been added in the Discussion section in the revised manuscript (Page 26, Lines 572-581). The detailed revisions are as follows:

"In addition, we must acknowledge that our study only captured the most important interactions in the basin. We cannot observe everything, everywhere, or all the time. Depicting all real-world processes is also challenging due to difficulties in mathematical assumptions and algorithm performance. Nevertheless, we believe that our findings are important for understanding the general watershed functioning and could guide the development of more accurate and region-specific eco-hydrological models. Models that are causally similar to observations (i.e., our causality results) may yield more reliable future projections (Runge et al., 2019). For example, in the area where snowmelt contributes significantly to runoff, a snowmelt module considering the accurate influencing time is

required in the model. In places where groundwater contributes greatly to the upper soil layers and the water uptake by roots, modules regarding groundwater and soil water movement should be considered carefully. We promote the use of network-based approaches and models together in the future to more formally address the perceptions of causality in hydrology and to better prepare for a broad range of possible futures."

(4) Occasional grammatical errors should be checked and corrected.

**Reply:** Thank you! We have carefully checked the text and correct the errors.

**Minor comments:**

(1) Lines 24-27 - It would be better to expand the introduction of eco-hydrological systems and internal interactions with more information.

**Reply:** As suggested, we have made more introduction to this and rephrased the corresponding paragraph.

The revised version can be found in the Introduction section of the revised manuscript (Pages 1-2, Lines 23-33). The detailed information are as follows:

"The hydrosphere and biosphere are intrinsically coupled subsystems of the Earth. Hydrological conditions shape the distribution, structure, and function of terrestrial ecosystems, which, in turn, affect the hydrological components via modulations of land-atmosphere water and energy fluxes (Pappas et al., 2017). Hence, eco-hydrological systems are complex with time-dependent interactions occurring between and within the atmosphere, vegetation, soil, and water bodies (Yan et al., 2023). These interactions contain intensifying and mitigating mechanisms, e.g., vegetation coverage can be enhanced by warmer temperatures, increased water availability, and afforestation, and can be further reduced by the decrease of water storage through root uptake. Together, these interactions among multiple components dictate a collective behavior of the eco-hydrological system (Goodwell et al., 2018). In the context of climate change and increasing human activities, eco-hydrological processes have undergone substantial changes. Therefore, it is a pressing need for a comprehensive understanding of how the system behaves (phenomenon) and unravelling the multivariate interactions (mechanisms) that drive such behaviors at the system level."

(2) Lines 85-87 - More emphasis should be placed on the reason for using the YRB as the study area.

**Reply:** Thanks for your comment. We use the Yellow River basin (YRB) in China as the study area due to the following reasons: (1) The YRB is an important ecological corridor, hosting more than 12% of population and creates about 14% of GDP of China. (2) The YRB has a vast area with different climatic conditions, land use types, and human disturbances, providing various types of eco-hydrological regimes for investigation. (3) The YRB has undergone significant changes in eco-hydrological processes due to climate change and intensive human activities. Hence, there is a need to investigate the exhibited evolution trends and the internal mechanisms in such eco-hydrological systems.

Related information has been clarified in the section of Introduction (Page 3-4, Lines 88-91). The detailed revisions are as follows:

"An important ecological corridor in China, the Yellow River Basin (YRB), which has been undergoing significant changes in eco-hydrological processes, is taken as the study case. The YRB is vast with different climatic conditions, land use types, and human disturbances, providing various types of eco-hydrological regimes for investigation (Luan et al., 2021; Wang et al., 2021; Yin et al., 2021)."

Related information has been clarified in the section of Study area (Page 10, Lines 249-251). The detailed revisions are as follows:

"The YRB is an important ecological corridor, hosting more than 12% of the population and creating about 14% of the GDP of China."

(3) Line 88 - A short presentation of the structure of the paper would be good here.

**Reply:** Thanks for the suggestion. A short presentation of the structure of the paper will be added in the last paragraph of Introduction.

Related information has been added in the last paragraph of Introduction (Page 4, Lines 93-96). The detailed revisions are as follows:

"The paper is structured as follows. Section 2 describes the framework developed. Section 3

introduces the study area and the data used. Section 4 presents the results for each subregion of the YRB, followed by a discussion of the findings in Section 5, including the significance of the study, comparisons with other studies, limitations, and future outlooks. Finally, some conclusions are drawn in Section 6."

(4) Lines 125-126 - I think the threshold has a significant influence on the construction of the network. How would the network and clustered modules change if you used a different threshold?

**Reply:** Yes, we fully agree with your comment that the threshold could influence the construction of the network. For comparison, Pearson's correlation coefficient (PCC)>0.4 and PCC>0.5 are also used as thresholds here. Although the existence of some links changes when different thresholds are used, the conclusions of the study remain unchanged.

The added information has been presented in the revised manuscript (Page 17, Lines 378-380). Corresponding figures have been added in the Supplementary Material (Figs. S2-S3). The detailed revisions are as follows:

"Given that the network structure and metrics can be influenced by using different thresholds, PCC>0.4 and PCC>0.5 were also employed to construct networks for validation (Figures S2 and S3)."

[Figure]

**Figure S2. Synchronous networks and corresponding clustered modules (when PCC>0.4).**

[Figure]

**Figure S3. Synchronous networks and corresponding clustered modules (when PCC>0.5).**

(5) Figure 2 - In terms of physically possible and plausible links, there is a connection between soil water storage (SMSA) and gross primary productivity (GPP), but not between soil water storage (SMSA) and normalized difference vegetation index (NDVI). Why is this?

**Reply:** In this study, NDVI represents vegetation coverage (i.e., the land surface condition) and GPP represents the physiological activity of vegetation. We assume the physically possible and plausible links based on Poppe Terán et al. (2023), i.e., enhanced photosynthesis (GPP) is directly supported by water supply from soil (SMSA) and contributes to the active growth of plants (NDVI). However, increased vegetation coverage (NDVI) in turn consumes water (SMSA) through physiological activity (GPP). Therefore, it is assumed that the relationship between NDVI and SMSA is mediated by the

variable GPP and is considered to be a spurious causal relationship.

The information above has been mentioned in the caption of Figure 2 in a brief way (Pages 8-9, Lines 200-202). The detailed revisions are as follows:

[Figure]

**Figure 2.** Overview of the causal inference method. (a) An example of causality that a lagged variable $X_{t-1}^4$ (the brown square) is said to be a cause of $X_t^1$ (the brown circle) if $X_{t-1}^4$ has a significant dependence or predictive power over $X_t^1$ while removing the effect of all other potential variables influencing $X_{t-1}^4$ or $X_t^1$ (the yellow squares), except $X_{t-1}^4$. (b) Four types of assumptions to construct physically possible and plausible links. $\tau_{max}$ means the maximum lag time. (c) The network with physically possible and plausible links between the included variables in the PCMCI analysis. PCMCI will test shown links for significant causality and yield the final causal network as a subset of this. The dashed line represents the causality considered to be spurious, but we do not remove it from the test as Case 4, as it might help illustrate eco-hydrological mechanisms in this study.

(6) Lines 201-202 - "PCMCI tests possible links and provides the final results as a subset of the total possible network……" However, in Figure 6, there are lines between SMSA and NDVI, as well as between GWSA and GPP (although you have defined them as spurious ones). It seems that the links beyond your hypothesis are also tested. Please check if the expression here is correct.

**Reply:** Thank you for making us notice. Our previous expression was inappropriate.

As stated in the reply to Question (5), we assume the link between SMSA and NDVI to be spurious due to GPP. Similarly, soil moisture affects the physiological activities of vegetation directly, and we think that groundwater usually affects vegetation by supplying water to the upper soil layers, so here the relationships between GWSA and GPP are assumed to be spurious too. However, we did not remove them from the causality test as "physically impossible" links, but left them in default status as we found such "spurious links" sometimes could be helpful in illustrating eco-hydrological mechanisms.

The relative description has been rewritten in the revised manuscript (Page 8, Lines 198-202). The detailed revisions are as follows:

"The network with physically possible and plausible links between the included variables in the PCMCI analysis. PCMCI will test shown links for significant causality and yield the final causal network as a subset of this. The dashed line represents the causality considered to be spurious, but we do not remove it from the test as Case 4, as it might help illustrate eco-hydrological mechanisms in this study."

(7) Section 4.1 - The ecohydrological conditions of eight subregions are not clear enough to me. This may hinder the understanding of the underlying mechanisms in the following sections. Apart from trends, I would recommend describing the average conditions of the subregions in brief.

**Reply:** Thanks for your suggestions. The description of average eco-hydrological conditions for each subregion have been added to Section 4.1, and some sentences have been rephrased to improve the readability.

Related information has been modified in the revised manuscript (Page 14, Lines 330-345). The detailed revisions are as follows:

"In the source regions (Regions I-II), the water resources were relatively abundant with high $R_{modulus}$, and most of the hydrological variables exhibited increasing trends (significant or insignificant) except for GWSA. The trend in snow cover area in the source region was not significant. However, the snow cover for melting (April) increased, and the onset of melting shifted earlier from June to May (Figure S1). In Regions III-VI on the Loess Plateau, $R_{modulus}$ became much lower compared to the source regions and showed a decreasing trend, except for Region VI (which is disturbed by the

reservoir). TWSA and GWSA all showed significant downward trends, with depletion increasing from upstream to downstream, while SMSA displayed non-significant upward trends. In the lower reaches (Region VIII), all the hydrological variables showed scarcity and declined substantially from 2003 to 2019. Regarding the regional sediment loads ($SL_{\_modulus}$), their evolution seemed to be irregular across the basin, with significant trends only in Regions VII (with XLD reservoir) and VIII (with severe water withdrawals).

Ecological conditions differed from hydrological conditions a lot. The poorest areas in terms of vegetation coverage (NDVI) and productivity (GPP) were the driest Regions III-IV, while for WUE the poorest part of the YRB was the source region where the temperature is low. NDVI and GPP of the growing season increased by 31.16% and 35.70% for the entire YRB, respectively. It indicated that the large-scale vegetation restoration undertaken over the last two decades was effective (Yu et al., 2023). However, the ecosystem water use efficiency (WUE) of the growing season decreased significantly in most subregions (except in Regions I and VIII) from 2003 to 2019."

(8) Figure 4 - "A gray box denotes no data ……" However, grey and blue are difficult to distinguish in Figures 4(d) and 4(h). In addition, Figure 4(h) lacks a ")", and the symbol "*" in Figure 4(i) is difficult to recognize.

**Reply:** Thank you for bringing the point to our attention. We have revised this figure in the revised manuscript.

The modification can be found in the revised manuscript (Page 15, Figure 4). The detailed revisions are as follows:

[Figure]

**Figure 4.** Eco-hydrological variables of the growing season, where the horizontal axis represents the year and the vertical axis is different subregions: (a) terrestrial water storage anomalies (TWSA); (b) soil water storage anomalies (SMSA); (c) groundwater storage anomalies (GWSA); (d) runoff increment modulus (R_modulus); (e) normalized difference vegetation index (NDVI); (f) gross primary productivity (GPP); (g) ecosystem water use efficiency (WUE); (h) sediment load increment modulus (SL_modulus); (i) Z statistic values of the M-K test for each eco-hydrological variable. The significance level is taken as 0.05. A gray box denotes no data; a red box represents a positive trend; a blue box represents a negative trend; the symbol * means the trend is significant.

(9) Figures 5 and 6 - The resolution needs to be enhanced.

**Reply:** We have enhanced the resolution of these figures.

The modification can be found in the revised manuscript (Page 16, Figure 5; Page 18, Figure 6). The detailed revisions are as follows:

[Figure]

**Figure 5.** (a) Correlation metrics for each subregion; (b) Module composition of positively correlated networks in different subregions. Different gray circles in the background represent different modules. Black lines represent correlations in the same module, and red lines represent correlations in different modules. Blue circles indicate variables of the hydrological subsystem, and green circles indicate variables of the ecological subsystem. WUE is a special ecological indicator represented in yellow

circles, as it is the coupling of hydrological (ET) and ecological (GPP) processes.

[Figure]

**Figure 6.** Causal process networks of eco-hydrological variables in the growing season (April to September) for Regions I-VIII. A link is only shown if found statistically significant at a 99% confidence level. Link labels in (1), (2) or (3) indicate the lag at which the connection is found, and only the strongest one is shown in the graph for clarity. (0) means a contemporaneous link and "—" indicates a contemporaneous link with uncertain direction. All links regarding WW are special, as they are determined by correlations, marked by PCC. Links between SMSA and NDVI as well as GWSA and GPP are regarded as spurious ones, which are denoted in dash lines. The red circle under P or (and) T indicates its dominance in controlling the local eco-hydrological system.

(10) Figure 6 - This figure is interesting and contains a large amount of information. To my best knowledge, the source region (subregion I) has frozen soil, yet temperature does not appear to significantly affect soil moisture. Could you explain this further?

**Reply:** Thank you for your comments. Seasonally frozen ground, sporadic permafrost, and predominantly continuous permafrost coexist in the source area of the YRB, and the spatial distribution of the frozen ground is diverse and complex (Song et al., 2024). However, it is difficult for us to obtain reliable frozen ground data, so the variable directly describing the frozen ground is not included in this study. The relationship between air temperature (T) and soil water storage (SMSA) may partially reflect the degradation of permafrost due to warmer climate and its potential impact on runoff. However, no significant causality between T and SMSA is captured in our case study, meaning that the increase in SMSA during the growing season (i.e. the thaw period) is mainly due to the contribution of precipitation (P). Similar results are found in Li et al. (2024). Meanwhile, previous studies have indicated that the effects of frozen ground degradation on soil moisture and runoff generally occur in winter in the source area of the YRB (Yang et al., 2023). The reduction in frozen ground depth has only a small positive effect on soil moisture in the growing season (Qin et al., 2017), because the soil ice content decreases and the soil liquid water content increases with increasing temperature, but this water can be quickly consumed by the evapotranspiration process.

This point has been added in the Discussion of the revised manuscript in a brief way (Page 22, Lines 491-494). The detailed revisions are as follows:

"P was discovered to dominate the evolutions of hydrological components in the source region, just as

Li et al. (2024) reported. However, increasing T had minor influences on the hydrological subsystem. This is due to the relatively small proportion of snow and glaciers (about 6% of the area; Table S2) and the insignificant contribution of the frozen-ground thawing process to soil moisture and runoff during the growing season (Qin et al., 2017; Yang et al., 2023)."

(11) Lines 355-359 - "Instead, increased T (Figure S3) was the dominant factor stimulating GPP……" "Meanwhile, increased P (Figure S3) was the crucial driver of the increases in the hydrological subsystem……" To interpret the mechanisms clearer, I prefer to present the temperature and precipitation time series (or trends) in the main body of the manuscript.

**Reply:** Thank you! In the main body of the revised manuscript, we have included the trends in P and T as suggested.

The modification can be found in the revised manuscript (Page 14, Figure 4). The detailed revisions are as follows:

[Figure]

**Figure 4.** Eco-hydrological variables of the growing season, where the horizontal axis represents the

year and the vertical axis is different subregions: (a) terrestrial water storage anomalies (TWSA); (b) soil water storage anomalies (SMSA); (c) groundwater storage anomalies (GWSA); (d) runoff increment modulus ($R_{modulus}$); (e) normalized difference vegetation index (NDVI); (f) gross primary productivity (GPP); (g) ecosystem water use efficiency (WUE); (h) sediment load increment modulus ($SL_{modulus}$); (i) Z statistic values of the M-K test for each eco-hydrological variable. The significance level is taken as 0.05. A gray box denotes no data; a red box represents a positive trend; a blue box represents a negative trend; the symbol * means the trend is significant.

(12) Lines 375-377 - I think "essentially" here is strange. The sentence needs to be rephrased.

**Reply:** Thank you! We have rewritten this sentence.

The modification can be found in the revised manuscript (Page 19, Lines 417-418). The detailed revisions are as follows:

"Similar links between NDVI and SMSA were detected, although they were treated as "spurious" ones (Section 2.3.3)."

(13) Line 420 - I think "modest" here is not appropriate. The sentence needs to be rephrased.

**Reply:** Thank you for the comment. We have rewritten this sentence.

The modification can be found in the revised manuscript (Page 21, Lines 456-457). The detailed revisions are as follows:

"Consequently, the decreasing trends of WUE in these two regions were relatively small."

**References:**

Bo, Y., Li, X.K., Liu, K., Wang, S.D., Zhang, H.Y., Gao, X.J., and Zhang, X.Y.: Three Decades of Gross Primary Production (GPP) in China: Variations, Trends, Attributions, and Prediction Inferred from Multiple Datasets and Time Series Modeling, Remote Sens., 14, 2564, https://doi.org/10.3390/rs14112564, 2022.

Bonotto, G., Peterson, T.J., Fowler, K., and Western, A.W.: Identifying Causal Interactions Between Groundwater and Streamflow Using Convergent Cross-Mapping, Water Resour. Res., 58, e2021WR030231, https://doi.org/10.1029/2021WR030231, 2022.

Cao, Y.P., Xie, Z.Y., Woodgate, W., Ma, X.L., Cleverly, J., Pang, Y.J., Qin, F., and Huete, A.: Ecohydrological decoupling of water storage and vegetation attributed to China's large-scale ecological restoration programs, J. Hydrol., 615, 128651, https://doi.org/10.1016/j.jhydrol.2022.128651, 2022.

Delforge, D., de Viron, O., Vanclooster, M., Van Camp, M., and Watlet, A.: Detecting hydrological connectivity using causal inference from time series: synthetic and real karstic case studies, Hydrol. Earth Syst. Sci., 26, 2181–2199, https://doi.org/10.5194/hess-26-2181-2022, 2022.

Ge, J., Pitman, A. J., Guo, W., Zan, B., and Fu, C.: Impact of revegetation of the Loess Plateau of China on the regional growing season water balance, Hydrol. Earth Syst. Sci., 24, 515-533, https://doi.org/10.5194/hess-24-515-2020, 2020.

Kelleher, C., McGlynn, B., and Wagener, T.: Characterizing and reducing equifinality by constraining a distributed catchment model with regional signatures, local observations, and process understanding, Hydrol. Earth Syst. Sci., 21, 3325-3352, https://doi.org/10.5194/hess-21-3325-2017, 2017.

Li, C.C., Zhang, Y.Q., Shen, Y.J., and Yu, Q.: Decadal water storage decrease driven by vegetation changes in the Yellow River Basin, Sci. Bull., 65, 1859-1861, https://doi.org/10.1016/j.scib.2020.07.020, 2020.

Li, X., Zhou, Z.H., Liu, J.J., Xu, C.Y., Xia, J.Q., Wang, P.X., Wang, H., and Jia, Y.W.: Analysis and partitioning of terrestrial water storage in the Yellow River source region, Hydrol. Process., 38, 2, https://doi.org/10.1002/hyp.15097, 2024.

Lv, M.X, Ma, Z.G., Li, M.X., and Zheng, Z.Y.: Quantitative analysis of terrestrial water storage changes under the grain for green program in the Yellow River Basin, J. Geophys. Res. Atmos., 124, 1336-1351, https://doi.org/10.1029/2018JD029113, 2019.

Poppe Terán, C., Naz, B.S., Graf, A. et al.: Rising water-use efficiency in European grasslands is driven by increased primary production, Commun. Earth Environ., 4, 95, https://doi.org/10.1038/s43247-023-00757-x, 2023.

Qin, Y., Yang, D.W., Gao, B., Wang, T.H., Chen, J.S., Chen, Y., Wang, Y.H., and Zheng, G.H.: Impacts of climate warming on the frozen ground and eco-hydrology in the Yellow River source region, China, Sci. Total Environ, 605-606, 830-841, https://doi.org/10.1016/j.scitotenv.2017.06.188, 2017.

Runge, J., Bathiany, S., Bollt, E., et al.: Inferring causation from time series in Earth system

sciences, Nat. Commun., 10, 2553, https://doi.org/10.1038/s41467-019-10105-3, 2019.

Song, L., Wang, L., Luo, D. et al.: Assessing hydrothermal changes in the upper Yellow River Basin amidst permafrost degradation, npj Clim. Atmos. Sci., 7, 57, https://doi.org/10.1038/s41612-024-00607-3, 2024.

Tursun, A., Xie, X., Wang, Y.B., Liu, Y., Peng, D.W., and Zheng, B.Y.: Enhancing streamflow simulation in large and human-regulated basins: Long short-term memory with multiscale attributes, J. Hydrol., 630, 130771, https://doi.org/10.1016/j.jhydrol.2024.130771, 2024.

Wang, T.H, Yang, H.B., Yang, D.W., Qin, Y., and Wang, Y.H.: Quantifying the streamflow response to frozen ground degradation in the source region of the Yellow River within the Budyko framework, J. Hydrol., 558, 301-313, https://doi.org/10.1016/j.jhydrol.2018.01.050, 2018b.

Wang, Z.J., Xu, M.Z., Penny, G., Hu, H.C., Zhang, X.P., and Tian, S.M.: Impact of revegetation and agricultural intensification on water storage variation in the Yellow River Basin, J. Hydrol., 635, 131218, https://doi.org/10.1016/j.jhydrol.2024.131218, 2024.

Yang, J.J., Wang, T.H., Yang, D.W., and Yang, Y.T.: Insights into runoff changes in the source region of Yellow River under frozen ground degradation, J. Hydrol., 617, 128892, https://doi.org/10.1016/j.jhydrol.2022.128892, 2023.

Zhang, B.Q., Tian, L., Yang, Y.T., and He, X.G.: Revegetation Does Not Decrease Water Yield in the Loess Plateau of China, Geophys. Res. Lett., 49, e2022GL098025, https://doi.org/10.1029/2022GL098025, 2022b.

Zhao, F.B., Ma, S., Wu, Y.P., Qiu, L.J., Wang, W.K., Lian, Y.Q., Chen, J., and Sivakumar, B.: The role of climate change and vegetation greening on evapotranspiration variation in the Yellow River Basin, China, Agric. For. Meteorol., 316, 108842, https://doi.org/10.1016/j.agrformet.2022.108842, 2022.